EMBO
Molecular Medicine

# CEP55 is a determinant of cell fate during perturbed mitosis in breast cancer

Murugan Kalimutho[1,2,*,†] (iD), Debottam Sinha[1,2,†], Jessie Jeffery[1], Katia Nones[1,3], Sriganesh Srihari[4], Winnie C Fernando[1], Pascal HG Duijf[5], Claire Vennin[6,7], Prahlad Raninga[1], Devathri Nanayakkara[1], Deepak Mittal[1], Jodi M Saunus[1,8], Sunil R Lakhani[8,9,10], J Alejandro López[1,2], Kevin J Spring[11,12,13], Paul Timpson[6,7], Brian Gabrielli[5,14], Nicola Waddell[1] & Kum Kum Khanna[1,**] (iD)

## Abstract

The centrosomal protein, CEP55, is a key regulator of cytokinesis, and its overexpression is linked to genomic instability, a hallmark of cancer. However, the mechanism by which it mediates genomic instability remains elusive. Here, we showed that CEP55 overexpression/knockdown impacts survival of aneuploid cells. Loss of CEP55 sensitizes breast cancer cells to anti-mitotic agents through premature CDK1/cyclin B activation and CDK1 caspase-dependent mitotic cell death. Further, we showed that CEP55 is a downstream effector of the MEK1/2-MYC axis. Blocking MEK1/2-PLK1 signaling therefore reduced outgrowth of basal-like syngeneic and human breast tumors in *in vivo* models. In conclusion, high CEP55 levels dictate cell fate during perturbed mitosis. Forced mitotic cell death by blocking MEK1/2-PLK1 represents a potential therapeutic strategy for MYC-CEP55-dependent basal-like, triple-negative breast cancers.

**Keywords** aneuploidy; breast cancer; centrosomal protein; CEP55; genomic instability

**Subject Categories** Cancer; Pharmacology & Drug Discovery

## Introduction

Centrosomal proteins have long been recognized as scaffold proteins, regulating both the mitotic spindle and microtubule organization, and hence are critical for cell cycle progression, reviewed in Kumar *et al* (2013). CEP55 (also known as *FLJ10540/C10orf3*) is a coiled-coil centrosomal protein originally identified as an indispensable regulator of cytokinesis, the final stage of cell division that results in the physical separation of two daughter cells (Fabbro *et al*, 2005; Martinez-Garay *et al*, 2006; Zhao *et al*, 2006; van der Horst & Khanna, 2009; van der Horst *et al*, 2009). CEP55 localizes to the centrosome throughout the cell cycle, to the mitotic spindle during mitosis and the midbody during cytokinesis (Lee *et al*, 2008). Cytokinesis is a tightly controlled process during cell division requiring multicomponent subunits that are recruited to the midbody in a CEP55-dependent manner (Fabbro *et al*, 2005; Bastos & Barr, 2010; Mondal *et al*, 2012; Agromayor & Martin-Serrano, 2013). These events are primarily mediated by the Endosomal Sorting Complex Required for Transport (ESCRT) machinery to ensure equal segregation of cytoplasmic contents between daughter cells (Lee *et al*, 2008). Failure to coordinate these events results in primary genetic lesions that promote genomic instability and tumorigenesis (Agromayor & Martin-Serrano, 2013).

CEP55 is expressed at low levels in most normal human tissues except testis, and its dysregulation is linked to multiple disease states (Sakai *et al*, 2006; Chen *et al*, 2009b; Inoda *et al*, 2009; Waseem *et al*, 2010; Janus *et al*, 2011; Shiraishi *et al*, 2011;

1   QIMR Berghofer Medical Research Institute, Herston, Qld, Australia
2   School of Natural Sciences, Griffith University, Nathan, Qld, Australia
3   Queensland Centre for Medical Genomics, Institute for Molecular Bioscience, The University of Queensland, St. Lucia, Qld, Australia
4   Computational Systems Biology Laboratory, Institute for Molecular Bioscience, The University of Queensland, St. Lucia, Qld, Australia
5   University of Queensland Diamantina Institute, Translational Research Institute, The University of Queensland, Brisbane, Qld, Australia
6   Cancer Division, Garvan Institute of Medical Research and The Kinghorn Cancer Centre, Sydney, NSW, Australia
7   Faculty of Medicine, St Vincent's Clinical School, University of NSW, Sydney, NSW, Australia
8   Centre for Clinical Research, The University of Queensland, Herston, Qld, Australia
9   School of Medicine, The University of Queensland, Herston, Qld, Australia
10  Pathology Queensland, The Royal Brisbane and Women's Hospital, Herston, Qld, Australia
11  Liverpool Clinical School, University of Western Sydney, Liverpool, NSW, Australia
12  Ingham Institute, Liverpool Hospital, Liverpool, NSW, Australia
13  South Western Sydney Clinical School, University of New South Wales, Liverpool, NSW, Australia
14  Mater Research Institute, Translational Research Institute, The University of Queensland, Brisbane, Qld, Australia
    *Corresponding author. Tel: +61 38453772; E-mail: murugan.kalimutho@qimrberghofer.edu.au
    **Corresponding author. Tel: +61 7 3362 0338; E-mail: kumkum.khanna@qimrberghofer.edu.au
    †These authors contributed equally to this work

Chen *et al*, 2012; Hwang *et al*, 2013; Jeffery *et al*, 2015; Zhang *et al*, 2016a; Bondeson *et al*, 2017; Frosk *et al*, 2017). It is associated with aggressive behavior in *in vivo* models, is an independent marker of poor clinical outcome in various malignancies, and has been recognized as a strong candidate for vaccine development in breast and colorectal cancers (Inoda *et al*, 2009, 2011a,b). In terms of its mechanistic involvement in cancers, CEP55 increases anchorage-independent growth, migration, and invasion *in vitro* and promotes tumor formation in nude mice, possibly through VEGFA-PI3K/AKT signaling (Chen *et al*, 2007; Inoda *et al*, 2009; Hwang *et al*, 2013). Gene expression studies implicated *CEP55* in progression from *in situ* to invasive breast cancer (Ma *et al*, 2003), and overexpression in primary breast tumors is a marker of chromosomal instability and poor prognosis (Carter *et al*, 2006; Fournier *et al*, 2006). Therefore, it appears that *CEP55* overexpression plays a pivotal role in tumorigenesis, likely through the emergence of aneuploidy. However, the mechanism of how CEP55 mediates genomic instability, aneuploidy, and tumorigenesis has remained elusive.

In this study, we provide the first experimental evidence directly linking CEP55-dependent aneuploidy to breast cancer survival. Using large breast datasets with clinical follow-up information, we confirmed that high levels of *CEP55* mRNA associate with poor clinical outcomes. Knockdown of *CEP55* in breast cancer cells *in vitro* significantly reduced the number of aneuploid cells, induced cell death during perturbed mitosis, and sensitized cells to anti-mitotic agents. Rapid onset of G2/M entry due to premature CDK1/cyclin B activation primed cell death following treatment with anti-mitotic agents in a CEP55-dependent manner. Furthermore, we found that CEP55 is a downstream effector of mitogen-activated protein kinase (MAPK)-MYC signaling. Dual inhibition of MAPK signaling (MEK1/2 inhibition) and the mitotic pathway (PLK1 inhibition) synergistically reduced the outgrowth of both murine and human breast cancer cells. These results provide a rationale for clinically targeting CEP55-dependent pathways in basal-like, triple-negative breast tumors for better treatment efficacy.

# Results

## CEP55 overexpression is associated with poor outcome in breast cancer

Although CEP55 is ubiquitously overexpressed in many human cancers (Jeffery *et al*, 2015), a detailed molecular understanding of its role in tumorigenesis has remained elusive. We analyzed *CEP55* expression using the publically available Gene expression-based Outcome for Breast cancer Online (GOBO) database (*n* = 1,881; Ringner *et al*, 2011). We found that *CEP55* mRNA expression is associated with the PAM50 breast cancer molecular subtypes (Luminal A, Luminal B, HER2, and basal-like), with the basal-like subtype exhibiting significantly higher expression of *CEP55* compared to other subtypes (*P* < 0.0001; Fig EV1A, available online). This increased expression of *CEP55* was also associated with high-grade tumors (*P* < 0.0001; Fig EV1B) and estrogen receptor (ER) status (*P* < 0.0001; Fig EV1C). *CEP55* high expression was significantly associated with poor overall survival (*P* = 0.00102), relapse-free survival (*P* < 0.00001), and distant metastasis-free survival

(*P* = 0.01135) (Fig EV1D–F). Similarly, a strong association between high CEP55 expressing tumors and poor survival was observed when we used a larger patient dataset from KMPlotter (*n* = 4,142; Appendix Fig S1A; Gyorffy *et al*, 2010). *CEP55* is part of a proliferation/mitotic gene signature suggesting that the observed differences in patient survival could be due to its association with proliferation. To rule out this possibility, we normalized the expression value of *CEP55* with key proliferation markers, *KI67* and *PCNA* using the TCGA (The Cancer Genome Atlas) dataset (*n* = 492) (Cancer Genome Atlas, 2012). This confirmed that *CEP55* expression was significantly higher in breast cancer patients compared to normal breast tissue independent of proliferation (*P* < 0.0001; Appendix Fig S1B and C). Collectively, these data provide compelling evidence that high expression of *CEP55* mRNA is associated with poor clinical outcomes in breast cancer and therefore could be a novel target for therapeutic intervention.

## Differential expression of CEP55 regulates breast cancer cell proliferation and survival

To help select suitable models for functional work, we first analyzed *CEP55* expression in a published breast cancer cell line gene expression array dataset (*n* = 51 lines; Neve *et al*, 2006). Similar to clinical samples, *CEP55* mRNA expression was higher in basal-like, triple-negative cell lines, particularly those with mesenchymal and invasive phenotypes (Appendix Fig S2A–C). Immunoblotting analysis showed a similar trend toward higher protein expression in basal-like lines (Fig 1A), but most striking was the higher expression observed in *HRAS*-transformed "AT" and "CA" derivatives of MCF10A compared to the near-normal (diploid) MCF10A parental cells (Fig 1B; Soule *et al*, 1990). To better understand the potential role of CEP55 in breast cancer, we transiently knockdown *CEP55* with pooled siRNAs in a panel of breast cancer lines and noticed significantly reduced viability of 6/8 basal and 4/9 luminal/HER2 cell lines with cutoff of 50% inhibition, irrespective of their baseline CEP55 expression (Figs 1C and EV2A). Moreover, knockdown of *CEP55* in two representative basal-like lines resulted in significant induction of cell death as evident by increased proportion of cells with sub-G1 DNA content (Fig EV2B).

To further evaluate the impact of CEP55 on cellular proliferation and survival, we knockdown *CEP55* in MDA-MB-231 cells (shows highest protein level in the panel of cell lines (Fig 1A) and exhibits invasive behavior *in vitro*/*vivo*) by stable transduction with doxycycline-inducible shRNAs that target different regions in the *CEP55* transcript (Appendix Fig S2D). The inducible knockdown system used in this study was "leaky", as evidenced by reduced CEP55 expression in both sh#2 and sh#8 polyclonal lines in the absence of doxycycline compared to scrambled shRNA-transfected cells (hereafter referred to as control) (Fig 1D and Appendix Fig S2D). Previously, we have shown that complete knockdown of *CEP55* using siRNAs in Hela cells resulted in cytokinesis failure leading to multinucleation (Fabbro *et al*, 2005). However, partially knockdown of *CEP55* in breast cancer cells using shRNAs (sh#2 or sh#8) showed no significant changes in the number of cells displaying cytokinesis failure although complete knockdown of CEP55 by growing cells continuously in presence of doxycycline over 30 days showed increased proportion of cells in G2/M phase with multinucleation concomitant with increased cell death (Fig EV2C), suggesting that

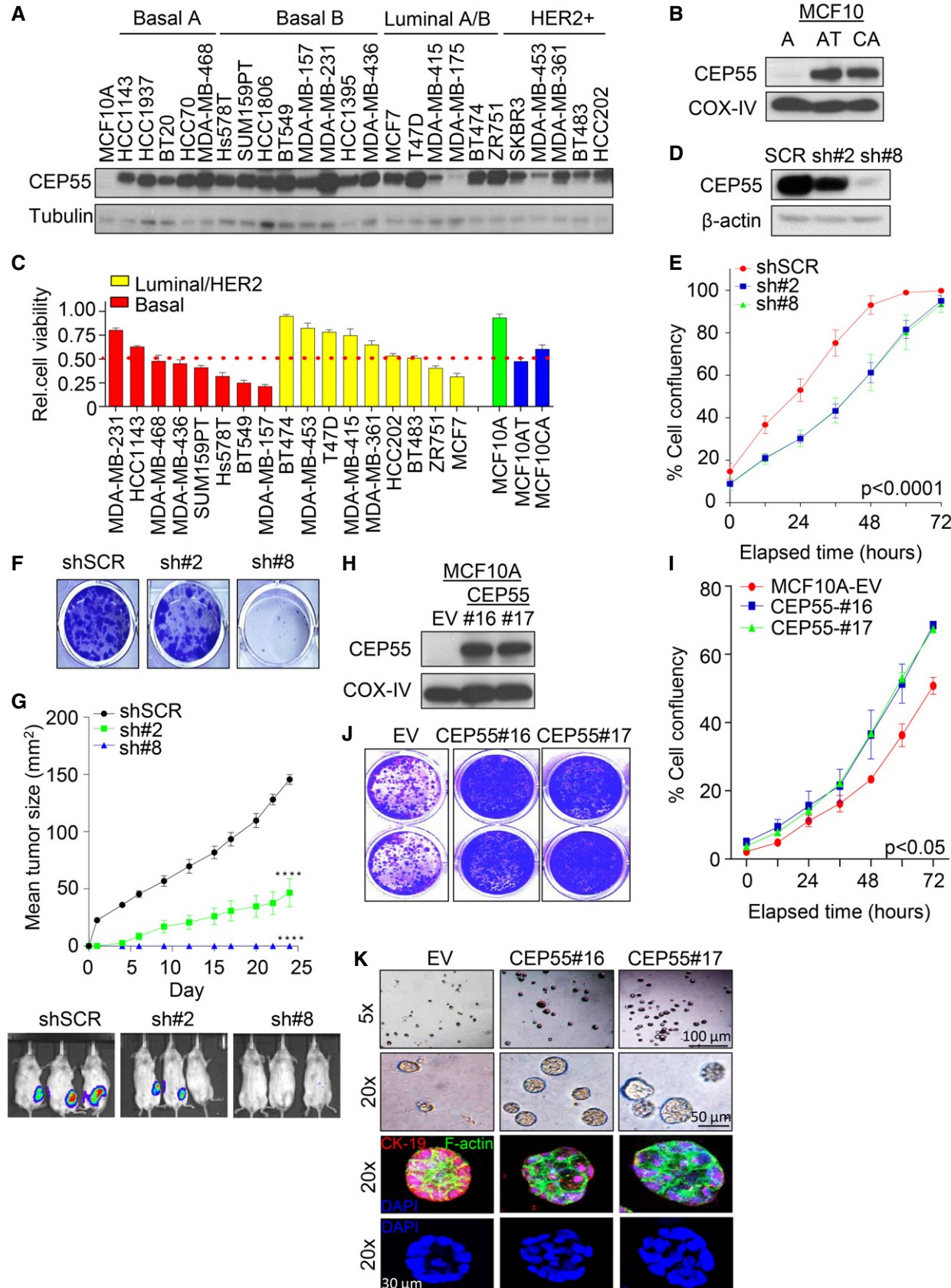

**Figure 1.**

◄

**Figure 1.  CEP55 regulates human breast cancer cell survival.**

A, B    Immunoblot analysis of CEP55 expression in a panel of human breast cancer lines (*n* = 25) and in progression series of MCF10A lines (MCF10AT and MCF10CA), respectively. Tubulin and COX-IV served as loading controls.

C        A panel of selected breast cancer and near-normal cell lines was reverse-transfected with 5 nM pooled CEP55 siRNAs and cell viability determined after 6 days. Cell viability relative to its own respective control transfected with scramble siRNA was calculated. Graph represents the mean ± SEM of three independent experiments.

D        Immunoblot analysis of CEP55 expression in CEP55 knockdown MDA-MB-231 cells. Two isogenic lines (sh#2 and sh#8) were obtained using two different shRNA sequences as described in method section. COX-IV served as loading control.

E        Effect of CEP55 knockdown on cell proliferation in MDA-MB-231 cells assessed using the IncuCyte ZOOM® live-cell imager (phase-only processing module). The percentage of cell confluence was determined using an IncuCyte mask analyzer. Graph represents the mean ± SEM of three independent experiments.

F        Representative images of colony-forming capacity at 14 days determined using crystal violet staining in control and CEP55 knockdown MDA-MB-231 cells.

G        Six-week-old female NOD/SCID cohorts of mice were injected in the 4th inguinal mammary fat pad with the control and CEP55 knockdown cells. Growth rate (area, mm$^2$) of the tumors was measured using digital calliper. Differences in growth were determined using Student's *t*-test, ****$P \leq 0.0001$. Graph represents the mean tumor area ± SEM, *n* = 6 mice/group.

H        Immunoblots analysis of CEP55 protein expression in MCF10A cells stably transfected with a Flag-CEP55 expression construct and two isogenic lines were obtained (CEP55-#16 and CEP55-#17) with its respective empty vector (EV) transfected parental cells. COX-IV served as a loading control.

I        Effect of CEP55 overexpression on cell proliferation in MCF10A assessed using the IncuCyte ZOOM® live-cell imager as described in panel (E). Graph represents the mean ± SEM of two independent experiments.

J        Representative images of colony-forming capacity at 14 days determined using crystal violet staining in empty vector (EV) and CEP55-overexpressing (CEP55#17 and CEP55#18) lines.

K        Representative images of single-plane phase-contrast and Z-stacked immunofluorescence of empty vector (EV) and CEP55-overexpressing MCF10A cells grown on Matrigel for 14 days. Red: cytokeratin 19; green: F-actin stained by Phalloidin and blue: DAPI. Z-stack images were acquired through Zeiss LSM 780 confocal microscope-ZMBH.

multinucleated cells are not viable over a long period of time. Therefore, stable CEP55 (sh#2 and sh#8) knockdown pools with reduced CEP55 expression (without doxycycline induction) were used in further studies to minimize the pronounced negative impact of inducible-complete CEP55 knockdown on cytokinesis failure and loss of viability. Both lines with reduced CEP55 exhibited significantly delayed proliferation which could be rescued by transiently expressed shRNA-resistant CEP55 expression construct ($P < 0.0001$; Figs 1E and EV2D and E). We also found that anchorage-independent colony formation was reduced upon CEP55 knockdown ($P < 0.0001$; Figs 1F and EV2F). Since CEP55 stimulates cell migration and invasion in oral and lung cancers (Chen *et al*, 2009a), we assayed *in vitro* migration and invasion in the MDA-MB-231 derivatives and observed a significant reduction compared to control cells ($P < 0.001$; Appendix Fig S3A and B). Engrafting a partial CEP55-knockdown MDA-MB-231 derivative (sh#2) showed a significant reduction in tumor growth while a near-complete CEP55 reduction (sh#8) abrogates tumor formation in NOD/SCID mice (Fig 1G and Appendix Fig S3C; 3 mice/group are shown as an example), suggesting that CEP55 is essential for tumor formation. To complement these knockdown experiments, we studied the effect of Flag-CEP55 overexpression in MCF10A cells, a near-normal mammary epithelial line, and generated two isogenic lines expressing high levels of CEP55 (Fig 1H, CEP55—#16 and—#17) (level of ectopic CEP55 is comparable to endogenous CEP55 found in highly aggressive MDA-MB-231 cells). Initial characterization of these cells demonstrated a significant increase in proliferation ($P < 0.01$; Fig 1I) and migration ($P < 0.0006$; Appendix Fig S3D).

Next, we tested whether the CEP55-overexpressing MCF10A cells had acquired an oncogenic phenotype by measuring their ability to form colonies in non-adherent conditions and form acinus-like structures on Matrigel. Both CEP55-overexpressing lines formed colonies more efficiently ($P < 0.0001$; Fig 1J, Appendix Fig S3E) and formed larger acinus-like spheroids on Matrigel compared to the empty vector (EV) transfected MCF10A cells (Fig 1K), and the latter are known to form growth-arrested structures with a well-defined

lumen in the center (Pal & Kleer, 2014). Moreover, we found that CEP55-overexpressing lines exhibited decreased staining of cytokeratin 19, a main cytoskeleton protein of epithelial cells and failed to form a hollow lumen as compared to control (EV) transfected MCF10A cells, suggesting that CEP55 overexpression is sufficient to confer malignant phenotype to near-normal MCF10A cells. Collectively, these data suggest that CEP55 overexpression confers anchorage-independent growth and a survival advantage to breast cancer cells.

**High levels of CEP55 are protective in aneuploid cells**

CEP55 is part of the 70-gene chromosomal instability (CIN) signature that was associated with aneuploidy in several human cancers (Carter *et al*, 2006); thus, we questioned whether high CEP55 expression could confer tolerance to or promote aneuploidy (causal effect). The first scenario would result in reduction of aneuploidy in CEP55-knockdown cells, and in the second scenario, CEP55 may act as a driver, increasing aneuploidy when overexpressed.

To test the first scenario, we performed cell cycle analysis by fluorescence-activated cell sorting (FACS) on the established CEP55-knockdown MDA-MB-231 surviving cells and found that the polyploid subpopulation was significantly reduced compared to the control cells ($P < 0.0001$; Fig 2A, yellow peaks). To confirm this, we performed metaphase spread analysis and found that both the mean and spread of the chromosome number were significantly reduced in sh#8 compared to control cells ($P < 0.0001$, *t*-test and $P < 0.0001$, *F*-test; Fig 2B). To confirm this was a *bona fide* effect and exclude the possibility of cell line or shRNA sequence-specific loss of aneuploidy, we knockdown *CEP55* in another well-characterized aneuploid line, Hs578T (Hackett *et al*, 1977), using constitutively expressed shRNA that target different region of *CEP55* transcript than those used for the MDA-MB-231 model (Fig EV3A). Analysis of cell proliferation and anchorage-independent growth showed a very similar pattern to the CEP55-knockdown MDA-MB-231 lines ($P < 0.0001$; Fig EV3B and C). Similarly, we found that

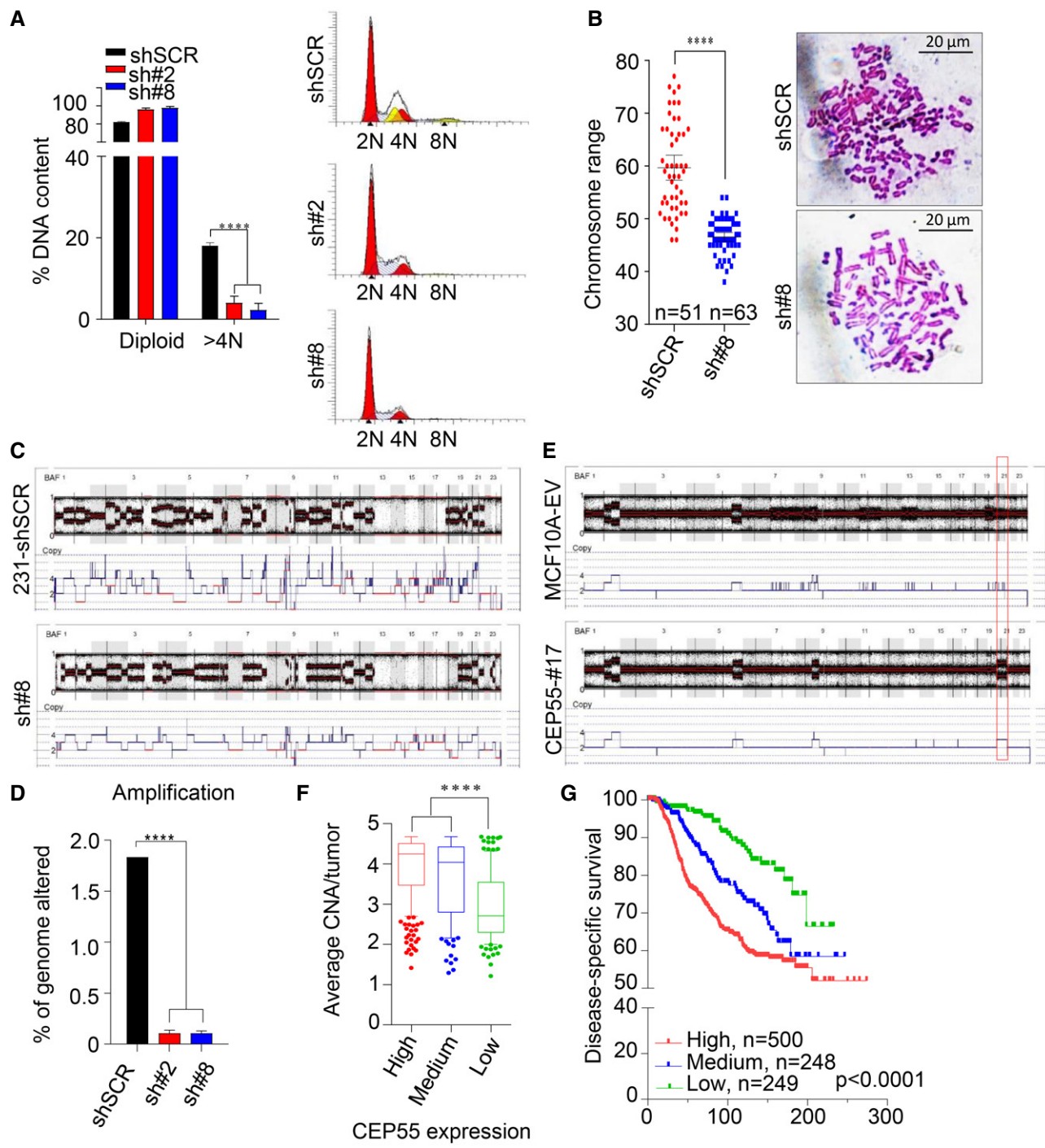

Figure 2.

aneuploid subpopulation was significantly reduced in *CEP55*-knockdown Hs578T cells compared to control ($P < 0.0001$; Fig EV3D), suggesting that the reduced aneuploid subpopulation was neither cell line- nor shRNA sequence-specific, but rather was a direct consequence of CEP55 loss.

We used single nucleotide polymorphism (SNP) array analysis to determine whether the copy-number profile was altered following CEP55 knockdown in MDA-MB-231 cells. Surprisingly, we observed a marked reduction in copy-number alterations

(CNAs) in the CEP55-knockdown cells (Figs 2C and EV3E). In particular, amplifications (defined by 6–8 copies) were significantly reduced ($P < 0.0001$; Fig 2D). To independently confirm these findings, we used the spontaneously metastatic MDA-MB-231-HM cell line derivative (HM-LNm5), which is highly genomically unstable with even more CNAs than the parental line (parental Fig 2C versus parental Fig EV3H and F). We observed a similar reduction of CNAs following CEP55 knockdown in these cells (Fig EV3G–I), further suggesting that CEP55 might be

**Figure 2. CEP55 regulates genomic instability and aneuploidy.**

A   Polyploidy analysis (> 4N DNA contents) determined using FACS in control and CEP55 knockdown MDA-MB-231 cells (left panel). Representative corresponding cytogram showing different phases of cell cycle and the polyploidy subpopulation analyzed using ModFit LT 4.0 software (right panel). Yellow peaks represent subpopulation of polyploidy/aneuploidy. Graph represents the mean $\pm$ SEM of three independent experiments. ****$P \leq 0.0001$.

B   Analysis of chromosome number in control and CEP55 knockdown MDA-MB-231 cells (sh#8) by Giemsa staining of metaphase spreads (left panel). Representative images showing chromosome spreads assessed by Zeiss AxioScope2 microscopy (right panel). $n$ = number of metaphase spreads counted (mean $\pm$ 95% confidence interval of two independent experiments). ****$P \leq 0.0001$.

C   Representative SNP array plots showing chromosomal alterations following CEP55-knockdown in MDA-MB-231 cells. BAF is B-allele frequency; value between 0 and 1 and represents the proportion contributed by one SNP allele (B) to the total copy number: BAF is an estimate of $N_B/(N_A + N_B)$, where $N_A$ and $N_B$ are the number of A and B alleles, respectively.

D   Percentage of genome altered evaluated using amplified copies (6–8 copies across genome) in both parental and CEP55 knockdown cells determined using data from SNP arrays. Graph represents the mean $\pm$ SEM of two independent experiments. ****$P \leq 0.0001$.

E   SNP array plots showing chromosomal alterations following CEP55 overexpression in MCF10A cells. Red box indicates an extra chromosome 20 in CEP55-overexpressing cells.

F   Correlation between average copy-number alterations of genome (CNAs)/tumor and *CEP55* mRNA expression in the METABRIC dataset, $n$ = 997 patients. ****$P \leq 0.0001$.

G   Kaplan–Meier survival analysis of METABRIC data showing the relationship between CEP55 expression and survival. Cases were subdivided into tertiles. Log-rank (Mantel–Cox) test, $P < 0.0001$.

required for survival of cells with genomic instability in heterogeneous populations.

Next, we tested whether overexpression of CEP55 is sufficient to induce CNAs in MCF10A cells stably overexpressing CEP55 (Fig 1H), by performing copy-number analysis using SNP arrays. We found an extra copy of chromosome 20 induced in these cells compared with empty vector-transfected cells (Fig 2E, red box). Ingenuity pathway analysis of the genes mapped to chromosome 20 (Fig EV3J) showed enrichment in cancer-associated processes (Fig EV3K) that may facilitate genomic instability and transformation in CEP55-overexpressing cells. The gain of only one chromosome in MCF10A cells stably overexpressing CEP55 was intriguing; therefore, we further tested the effect of transient overexpression of CEP55 in MCF10A cells. Notably, we found induction of polyploidy (defined here as cells with DNA content > 4N) in cells that transiently overexpress CEP55 (Fig EV3L). Taken together, the data suggest that in stable transduction experiments CEP55 may initially induce polyploidy, but those cells showing high levels of polyploidy may be selected against the cells with advantageous alterations like chromosome 20 amplification during long-term culture.

To investigate the potential clinical significance of these observations, we analyzed CNAs and expression data for breast cancers from the TCGA. Gains at chromosome 20 are very common in breast cancer, particularly the 20q arm (Hodgson *et al*, 2003; Tabach *et al*, 2011). Using TCGA data, we found that 20q is significantly more frequently gained than lost in breast tumor samples (15.6 versus

1.1%, $P = 1.16 \times 10^{-34}$; Appendix Fig S4A) and tumors with 20q gain are significantly associated with higher *CEP55* expression ($P < 0.0001$; Appendix Fig S4B). Moreover, we found that high *CEP55* expression is associated with high average copy number of genome in the METABRIC Discovery dataset ($n$ = 995 tumors; $P < 0.0001$; Fig 2F). This was consistent with clinical outcomes in these subgroups; breast cancer-specific survival in the *CEP55*-high, CNA-high group was significantly shorter than cases with lower *CEP55* expression and fewer CNAs ($P < 0.0001$; Fig 2G). Overall, the data suggest that high *CEP55* expression is associated with genomic instability.

## CEP55 overexpression provides survival advantage during perturbed mitosis

Next, we investigated mechanisms by which CEP55 overexpression could promote genomic instability. Since CEP55 is a microtubule-associated centrosomal protein that efficiently bundles microtubules (Zhao *et al*, 2006) and CEP55-knockdown cells exhibit reduced proliferation (Fig 1E and I), we challenged CEP55-knockdown cells with the microtubule-depolymerizing drug nocodazole or the PLK1 inhibitor (PLK1i) BI2536, which blocks cells in mitosis by inhibiting signaling cascades necessary for centrosome separation, microtubule–kinetochore attachments and normal mitotic progression. Notably, mitotic arrest with either agent was more efficient in CEP55-knockdown MDA-MB-231 cells compared to the control-treated

**Figure 3. CEP55 overexpression provides survival advantage during perturbed mitosis.**

A   Representative cytogram of control and CEP55 knockdown MDA-MB-231 cells showing cell cycle profiles following treatment with and without the BI2536 (5 nM).

B, C   Percentage of polyploidy (> 4N) following BI2536 (5 nM) or nocodazole (0.5 μM) in control and CEP55 knockdown MDA-MB-231 cells. Graph represents the mean $\pm$ SEM of three independent experiments. ns $P > 0.05$; ***$P \leq 0.001$; ****$P \leq 0.0001$.

D, E   Apoptotic fraction (sub-G1 population) determined by propidium iodide staining of cells treated with PLK1i or nocodazole as described in panels (B, C). Graph represents the mean $\pm$ SEM of three independent experiments. **$P \leq 0.01$; ****$P \leq 0.0001$.

F   Percentage of sub-G1 fraction following BI2536 (10 nM) or nocodazole (0.5 μM) in control and CEP55 knockdown Hs578T cells. Graph represents the mean $\pm$ SEM of two independent experiments. ns $P > 0.05$; *$P \leq 0.05$; **$P \leq 0.01$, ***$P \leq 0.001$.

G   Representative cytogram of empty vector (EV) and CEP55-overexpressing MCF10A cells showing cell cycle profile following treatment with and without the BI2536 (10 nM).

H   Percentage of polyploidy following treatment with and without BI2536 (10 nM) in empty vector (EV) and CEP55-overexpressing MCF10A cells. Graph represents the mean $\pm$ SEM of three independent experiments. *$P \leq 0.05$; ****$P \leq 0.0001$.

I   Corresponding sub-G1 population determined using propidium iodide staining of cells treated with BI2536 as described in panel (H). Graph represents the mean $\pm$ SEM of three independent experiments. *$P \leq 0.05$; **$P \leq 0.01$; ***$P \leq 0.001$.

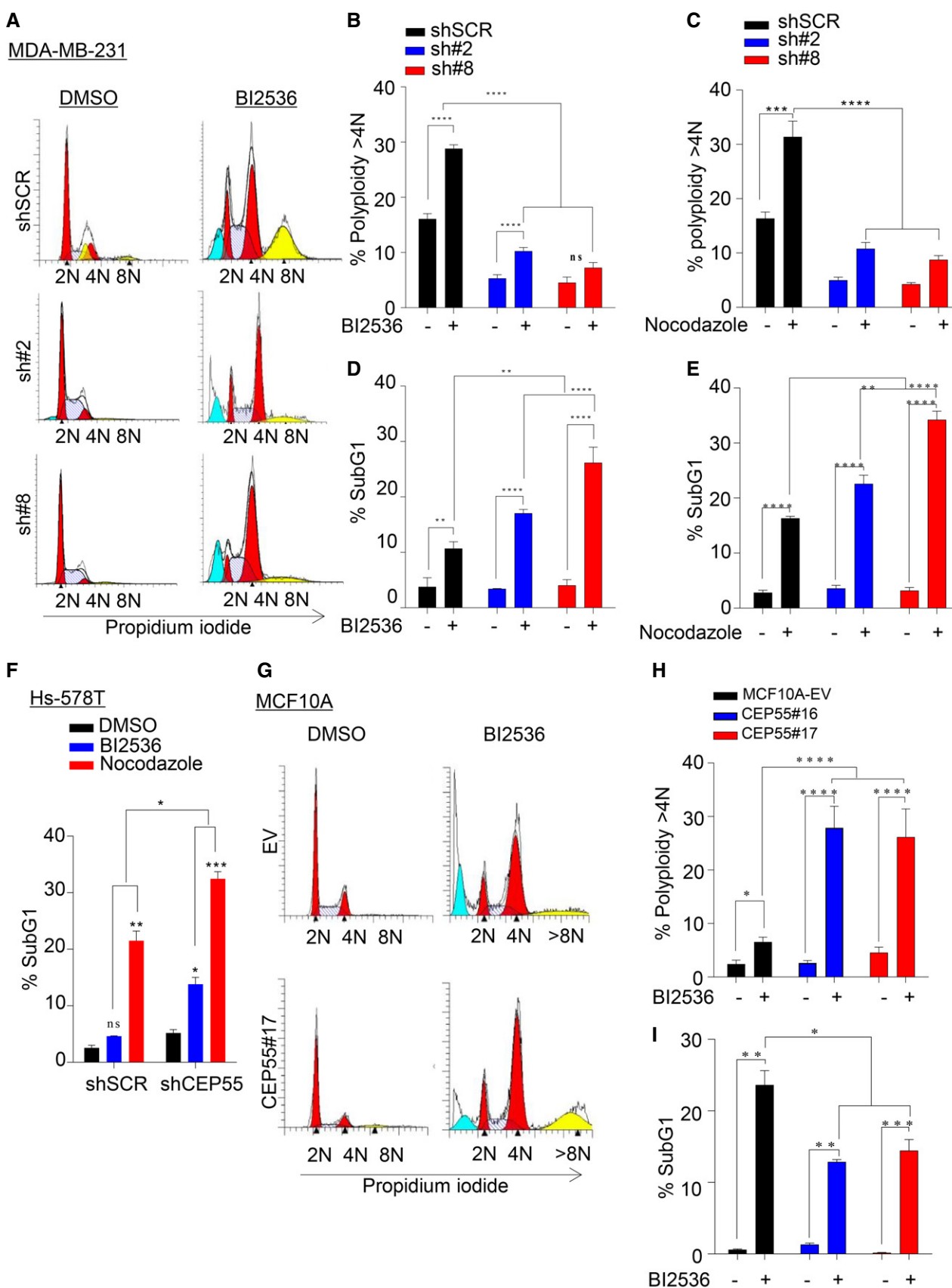

Figure 3.

cells (Fig 3A, data not shown). We noticed an increase in polyploidy following both nocodazole and PLK1i in control MDA-MB-231 cells, but lesser in CEP55-knockdown cells (Fig 3B and C), suggesting that CEP55 overexpression may facilitate premature exit during mitotic arrest by impairing cell death induced by these agents. Indeed, CEP55-knockdown MDA-MB-231 cells showed a significant increase in cell death (identified as the subG1 fraction) during perturbed mitosis, which could prevent the generation of polyploidy (Fig 3D and E). Similar results were obtained in CEP55-knockdown Hs578T cells treated with nocodazole and PLK1i (Fig 3F). Conversely, to determine whether CEP55 overexpression is sufficient to promote survival during perturbed mitosis, we treated empty vector (EV) and CEP55-overexpressing MCF10A cells with PLK1i. This significantly increased the polyploid cell fraction (Fig 3G and H) concomitant with significant reduction in the sub-G1 population in CEP55-overexpressing cells (Fig 3I), suggesting that overexpression of CEP55 blocked cell death during perturbed mitosis.

## CEP55 dictates cell fate following anti-mitotic drug treatment

The data presented above provides evidence that CEP55-knockdown cells cannot survive during perturbed mitosis and instead undergo apoptosis. On the contrary, CEP55-overexpressing cells under similar conditions are able to survive possibly in part by impairing cell death signals. To examine specifically whether anti-mitotic drug-induced apoptosis in CEP55-knockdown cells occurs during mitosis, we followed their fates using time-lapse microscopy after PLK1i treatment. There was no significant baseline difference in the average time spent in mitosis following CEP55 knockdown in growing cell culture (Appendix Fig S5A), but we found that upon PLK1 inhibition control MDA-MB-231 cells indeed spent more time in mitosis (Fig 4A). This behavior correlated with cell fate, as PLK1i-treated parental cells exhibited prolonged mitotic arrest and underwent

slippage, while a proportion of CEP55-knockdown cells died in mitosis and fewer cells slipped from mitotic arrest (P < 0.0001; Fig 4B, Appendix Fig S5B–D, Movie EV1–EV3).

To study whether cells were dying by caspase-dependent apoptosis, we pretreated them with the pan-caspase inhibitor Z-VAD and found that this rescued the phenotype (P < 0.0001; Fig 4C). Moreover, caspase inhibition prolonged mitotic arrest and shifted the fate profiles, reducing the number of cells that died in mitosis in favor of cells that prematurely exited, similar to the PLK1i-treated control cells (Fig 4D). This indicates that the mitotic cell death observed with CEP55 knockdown and PLK1i treatment was caspase-dependent. Conversely, when we repeated this experiment in CEP55-over-expressing MCF10A cells, mitotic arrest was prolonged compared to vector only transfected cells, with a significant increase in slippage and a reduction in cell death mimicking the phenotype of MDA-MB-231 cells expressing high levels of endogenous CEP55 (Fig EV4A and B). Collectively, these data suggest that overexpression of CEP55 blocks apoptosis and allows cell slippage.

Having established that CEP55-knockdown cancer cells were more sensitive to anti-mitotic agents and susceptible to apoptosis, we wanted to determine which of the apoptotic effectors are involved in this response. It is well documented that anti-apoptotic effectors like MCL-1, BCL-XL, and BCL-2 are tightly regulated during mitosis and are determinants of cell fate during perturbed mitosis (Topham & Taylor, 2013). We synchronized cells by double-thymidine block, released them into nocodazole or PLKi, and then performed immunoblotting for candidates. PARP and caspase-3 cleavage were increased in CEP55-knockdown cells treated with nocodazole or PLKi in a manner partially rescued by shRNA-resistant CEP55 expression construct (Fig EV4C and D). Moreover, we found that MCL-1 and BCL-2 did not show any correlation with nocodazole treatment across the cell lines tested, but the BCL-XL levels were suppressed following nocodazole treatment in CEP55-knockdown cells in a manner rescued by shRNA-resistant CEP55

**Figure 4. Overexpression of CEP55 rescues cell death during perturbed mitosis.**

A, B  Box and whiskers plot (A) showing average time spent in mitosis and (B) mitotic outcomes in control and CEP55 knockdown MDA-MB-231 cells following treatment with the BI2536 (5 nM). Time taken to complete mitosis was defined as the time from nuclear envelope breakdown until two daughter cells were observed whereas mitotic slippage or death was defined as cells that prematurely exited mitosis with a flattened and a multinucleated phenotype or died during mitosis, characterized by membrane blebbing. Graph represents the mean ± SEM of two independent experiments. For each experiments, $n$ = 50 mitotic cells were counted per condition using Olympus Xcellence IX81 time-lapse microscopy. ****$P \le 0.0001$.

C, D  Cells were pretreated for 2 h with 50 μM of the pan-caspase inhibitor Z-VAD-FMK, followed by treatment with BI2536 (5 nM), before assessment of cell fate (C) and an average time spent in mitosis (D) as described in panels (A and B). Graph represents the mean ± SEM of two independent experiments. ****$P \le 0.0001$.

E  Cells were synchronized using double-thymidine then released into nocodazole (0.5 μM), and protein lysates were collected at the indicated time points. Immunoblot analysis was then performed to determine the expression and activity of mitotic regulators as indicated. Levels of phospho-MEK$^{T286}$ and dephosphorylation of phospho-CDK1$^{Y15}$ served as markers of Cdk1 activation/mitotic entry. COX-IV served as a loading control.

F  Schematic representation of the mCerulean-CDK1-FRET biosensor. CDK1 activity is low when FRET signaling is low (high lifetime reading) versus CDK1 activity is high when the FRET signaling is high (low lifetime reading). Adapted from Vennin et al. (2017), reprinted with permission from AAAS.

G  Representative mCerulean lifetimes maps of Hs578T control and CEP55 knockdown cells upon nocodazole (0.5 μM) treatment at indicated time points. Cells were synchronized using double-thymidine for 16 h prior to nocodazole treatment.

H  Quantification of mCerulean lifetimes of control and Hs578T CEP55 knockdown cells and in response to treatment with nocodazole at indicated time points as shown in (G). Blue: 1 ns; red: 3 ns. Graph represents the mean ± SEM of three independent experiments. ns$P$ > 0.05; **$P \le 0.01$; ****$P \le 0.0001$.

I  Control and CEP55 knockdown MDA-MB-231 cells were synchronized as above, released into culture medium for 6 h and then blocked with nocodazole (0.5 μM) for an additional 4 h prior to treatment with the Cdk1 inhibitor RO-3306 for an additional 16 h. Sub-G1 cells were then identified by propidium iodide staining and quantified by FACS. Graph represents the mean ± SEM of two independent experiments. ****$P \le 0.0001$.

J  Control and CEP55 knockdown MDA-MB-231 cells were transfected with MAD2 siRNA (5 nM) for 48 h followed by nocodazole (0.5 μM) treatment for an additional 12 h. Cells were collected and fixed, and sub-G1 analysis was performed as described in methods. Graph represents the mean ± SEM of two independent experiments. ****$P \le 0.0001$.

Source data are available online for this figure.

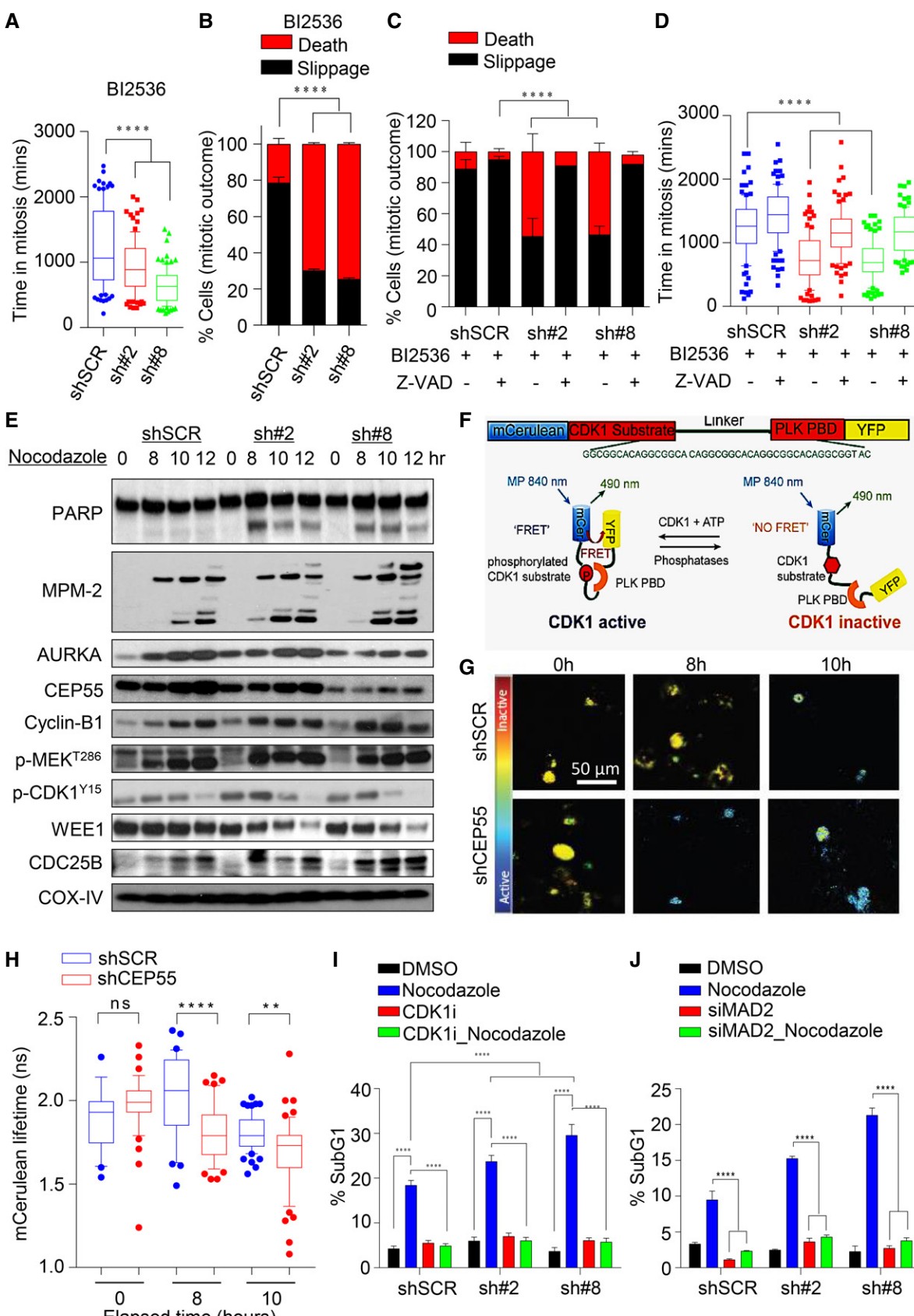

**Figure 4.**

construct (Fig EV4D). Furthermore, the baseline level of pro-apoptotic BH3-only protein BIM was higher in CEP55-knockdown cells and upon treatment the reduction in BIM levels was more pronounced in control cells compared to CEP55 knockdown cells (Fig EV4C). To address the role of BIM and BCL-XL in regulating apoptotic signaling in our model, we transiently knocked down either BIM or BCL-XL and analyzed cell death after nocodazole treatment. BIM knockdown failed to rescue the apoptosis in CEP55-knockdown cells (Fig EV4E). However, consistent with previous reports, we found that knockdown of BCL-XL resulted in enhanced nocodazole-induced apoptosis in control MDA-MB-231 cells, clearly indicating a role for BCL-XL in regulating this process (Bah *et al*, 2014; Bennett *et al*, 2016). Likewise, when we tested the effect of nocodazole treatment on CEP55 knockdown Hs578T cells, we found decreased levels of BCL-2, MCL-1, and BCL-XL (Fig EV4F) and baseline increase in BIM levels suggesting that CEP55-mediated regulation of apoptosis may also involve other Bcl-2 family members. Thus, apoptosis in this system may therefore reflect a more global dysregulation of Bcl-2 family members, which will be subject of further investigation. Conversely, when we used CEP55 overexpressing MCF10A cells which show reduced nocodazole-induced apoptosis (similar to other data, Fig 3I and Appendix Fig S6L), we found an increase in baseline expression of BCL-XL (Fig EV4G). These data cumulatively suggest that CEP55 loss may enhance death in mitosis after anti-mitotic treatment through regulation of Bcl-2 family members.

### Loss of CEP55 primes premature CDK1/cyclin B activation and apoptosis in the presence of anti-mitotic drugs

Several studies have demonstrated that premature CDK1/cyclin B activation is required for anti-mitotic drug efficacy (Yu *et al*, 1998; Castedo *et al*, 2002). To test the hypothesis that this also occurs in our models, we first measured the kinetics of G2/M entry in both control and CEP55-knockdown MDA-MB-231 cells. Cells were synchronized by double-thymidine block, released into culture medium, and collected at 2-h intervals for DNA content analysis. Notably, there was no obvious difference in the kinetics of G2/M entry, suggesting the CEP55-knockdown cells probably cycling in a similar manner to control cells (Appendix Fig S6A) and spending an equal average time in mitosis (Appendix Fig S5A). However, when synchronized cells were released in the presence of nocodazole or PLKi, the CEP55-knockdown cells cycled faster (Appendix Fig S6B and C) and showed significantly increased cell death (Appendix Fig S6D and E). Similar results were obtained using time-lapse microscopy when we followed the synchronized cells released into nocodazole, showing that CEP55 knockdown cells enter mitosis faster than control cells (Appendix Fig S6F), on average spend less time in mitosis (Appendix Fig S6G) and a fraction of cells died instead of undergoing slippage (Appendix Fig S6H). This phenotype was rescued by transient overexpression of shRNA-resistant CEP55 (Appendix Fig S6F–H). Similarly, biochemical analysis revealed an early induction of markers of G2/M entry, including cyclin B1, CDC25B, and pMEK1$^{T286}$ (a marker of CDK1/cyclin B activation; Fig 4E), Wee1 destabilization and dephosphorylation of CDK1$^{Y15}$, as well as accumulation of phospho-Ser/Thr-Pro MPM-2 in the CEP55-knockdown cells compared to control cells. Similar data were also obtained in CEP55-knockdown Hs578T cells (Appendix Fig S6I).

Conversely, CEP55-overexpressing MCF10A lines exhibited significantly delayed mitotic entry upon nocodazole treatment with reduced apoptosis upon mitotic arrest (Appendix Fig S6J–M) compared to empty vector-transfected control cells.

In addition, CDK1 activity was monitored using a CDK1-FRET biosensor activity probe as described previously (Gavet & Pines, 2010; Vennin *et al*, 2017). CDK1 activity is low when FRET signal is low (high mCerulean lifetime) versus CDK1 activity is high when FRET signal is high (low mCerulean lifetime) (Fig 4F). We found significantly lower mCerulean lifetime, indicating higher CDK1 activity within 8–10 h post-release of synchronized CEP55-knockdown Hs578T cells into nocodazole ($P < 0.001$, Fig 4G and H), suggesting premature entry of these cells in mitosis. Taken together, our data suggest that increased sensitivity of CEP55-knockdown cancer cells to anti-mitotic agents can be explained by faster entry into mitosis due to premature activation of CDK1.

Next, we asked whether inhibition of premature CDK1 activation in CEP55-knockdown cells could selectively rescue apoptosis induced by nocodazole. Synchronized cells were released into culture medium for 6 h and then treated with nocodazole prior to CDK1 inhibition (at which point the cells are already in mitosis). Addition of the CDK1 inhibitor RO-3306 significantly inhibited mitotic cell death induced by nocodazole in CEP55-knockdown cells (Fig 4I), suggesting that nocodazole-induced apoptosis was CDK1-dependent. Furthermore, silencing of the spindle assembly checkpoint (SAC) to breach cyclin B1 degradation-using siRNA against MAD2 (a core component of SAC) significantly blocked nocodazole-induced apoptosis (Fig 4J). Collectively, our data suggest that in the absence of CEP55, treatment with anti-mitotic agents induces an apoptotic threshold breach through premature CDK1 activation and SAC-dependent mitotic cell death.

### CEP55 overexpression mediates resistance to docetaxel-induced apoptosis

Considering that premature exit by impairing death signaling cascades have been implicated in resistance to microtubule poisons (Manchado *et al*, 2012), we questioned whether high expression of CEP55 could also contribute to chemotherapeutic resistance, particularly to docetaxel, which is widely used in breast cancer management. Accordingly, we challenged the MDA-MB-231 cells with sublethal doses of docetaxel and doxorubicin and indeed found that CEP55-knockdown cells were more susceptible to apoptosis with docetaxel but not doxorubicin (Fig EV5A). By monitoring cells treated with low concentration of docetaxel over 120 h, we found that CEP55-knockdown slowed proliferation (Fig EV5B). Similarly, when these cells were treated with an even lower dose of docetaxel (0.5 nM), control and sh#2 cells were not responsive, but sh#8 cells, with near-complete CEP55 knockdown, responded to this treatment with significantly reduced proliferation ($P < 0.0001$; Fig EV5C).

We also found a significant reduction in colony formation in CEP55-knockdown cells following 0.5 nM docetaxel treatment ($P < 0.0001$, Fig EV5D), further suggesting that CEP55 may contribute to anti-mitotic drug resistance. To investigate this in a more clinically relevant context, we analyzed the relationship between CEP55 expression and survival in breast cancer patients treated with chemotherapy (variable treatments) using KMPlotter datasets. Notably, *CEP55* mRNA expression was inversely associated with

relapse-free survival [HR: 1.67 (1.11–2.51); $n = 425$; log-rank $P = 0.013$] with a trend in poor overall survival [HR: 2.12 (0.81–5.59); $n = 69$; log-rank $P = 0.12$, Fig EV5E and F]. This suggests that CEP55 may contribute to chemotherapy resistance, and raises the possibility that it could be integrated into algorithms used to predict chemotherapy response or targeted as a chemotherapy sensitization strategy.

**MAPK pathway controls CEP55 levels through MYC**

The CEP55 protein structure features multiple coiled-coil domains that are notoriously undruggable, and so, we speculated that indirectly targeting upstream or downstream effectors may be a more effective way to manipulate CEP55 levels. Treating MDA-MB-231 cells with a number of small molecule inhibitors of EGFR/HER2 signaling revealed that CEP55 protein was markedly suppressed following inhibition of MEK1/2 and to a lesser extent with AKT and PI3K/mTOR, but not with upstream receptor tyrosine kinases (RTKs) EGFR, HER2, VEGFR, or PDGFR (Fig 5A). The suppression of CEP55 by MEK1/2 inhibitors occurred at both the mRNA and protein level (Fig 5B and Appendix Fig S7A) and was dose- and time-dependent (Fig 5B and Appendix Fig S7B), and this was not due to G1 cell cycle arrest (Appendix Fig S7C). Moreover, the promoter activity of *CEP55* was also significantly reduced with MEK1/2 inhibitor treatment or ERK1/2 silencing (Fig 5C, Appendix Fig S7D). In addition, EGF stimulation of growth-arrested MDA-MB-231 cells (in 0.1% fetal calf serum) markedly increased *CEP55* and *MYC* mRNA and protein levels (Fig 5D, Appendix Fig S7E), suggesting that MAPK signaling regulates CEP55 levels.

Upon mitogenic stimulation, MAPK signaling activates an array of downstream effectors including the transcription factor MYC, which is associated with genomic instability and progression in multiple cancers (Sears *et al*, 2000; Prochownik, 2008), and is known to be targeted by MEK1/2 blockade in breast cancer cells (Duncan *et al*, 2012). Therefore, we wondered whether MYC regulates CEP55 expression. SiRNA-mediated knockdown of *MYC* in MDA-MB-231 cells reduced *CEP55* promoter activity, mRNA, and protein levels with and without prior EGF stimulation of cells (Fig 5E and F, Appendix Fig S7F), while knockdown of *ETS-1* [another transcription factor downstream of MAPK (Ohtani *et al*, 2001; Foulds *et al*, 2004)] had no effect on CEP55 expression (Fig 5F, Appendix Fig S7F). Moreover, exogenous activation of *MYC* with 4-hydroxytamoxifen (4OHT) in MCF10A-MYCER cells (Sato *et al*, 2015) markedly increased CEP55 levels (Fig 5G and H). Finally, in the TCGA dataset, *CEP55* mRNA levels correlated with predicted MYC activity ($P < 0.0001$; Fig 5I). Collectively these data suggest that MAPK controls *CEP55* expression through MYC oncogene.

**CEP55 expression determines sensitivity to combined treatment of MEK1/2 and PLK1**

Though several lines of evidence suggest that CEP55 could be a candidate target for therapeutic development, there is currently no specific way to target CEP55 overexpressing cancers. First, CEP55 is a cancer–testis antigen, with expression restricted to testis in healthy adults; however, it is re-expressed in malignant cells in some cancer patients (Morita *et al*, 2007; Jeffery *et al*, 2015). Second, high CEP55 expression is an indicator of chromosomal instability (Carter *et al*, 2006) and poor clinical outcome (Jeffery *et al*, 2015). Evidence presented here shows this is likely because it promotes genomic instability and resistance to anti-mitotic drugs—even partial knockdown of CEP55 was sufficient for aneuploid breast cancer cell elimination *in vitro* (Figs 2–4). Since MEK1/2 inhibitors target *CEP55* through MYC (Fig 5B) and increased sensitivity of CEP55

---

**Figure 5.  CEP55 is a downstream effector of MAPK signaling.**

A  Immunoblots analysis was performed to determine CEP55 levels in MDA-MB-231 cells treated with multiple inhibitors targeting the EGFR/HER2 pathway for 24 h. The following inhibitors were used: MEK1/2i (AZD6244 (1 μM)), the AKT, PI3K/mTORi (BEZ235 (0.5 μM), AKTi VIII (1 μM)), EGFRi and HER2i (erlotinib (1 μM), afatinib (0.25 μM), lapatinib (1 μM), trastuzumab (10 μg/ml), or the pan VEGFR, PDGFR, and RAF kinases (sorafenib (1 μM)). COX-IV was used as a loading control.

B  Immunoblots analysis of MDA-MB-231 cells treated with three different MEK1/2 inhibitors at various concentrations (selumetinib AZD6244, trametinib GSK1120212, and binimetinib MEK162) after 24 h. COX-IV was used as a loading control.

C  Relative *CEP55* promoter luciferase activity in MDA-MB-231 cells either treated with AZD6244 (1 μM) for 6 h or ERK1/2 knockdown for 24 h. PGL basic reporter plasmid was used to normalize basal *CEP55* promoter activity. Graph represents the mean ± SEM of two independent experiments. *$P \leq 0.05$.

D  Immunoblot analysis of MDA-MB-231 cells cultured in 0.1% fetal bovine serum for 24 h and stimulated with EGF (100 ng/mL) for the indicated time points. Phosphorylation of EGFR, ERK1/2 along with CEP55, and MYC were determined. COX-IV as a loading control.

E  Relative *CEP55* promoter luciferase activity upon MYC siRNA was determined using Dual-Glo assay in MDA-MB-231 cells similar to experiment in panel (C). Graph represents the mean ± SEM of two independent experiments. *$P \leq 0.05$.

F  Immunoblots analysis showing MYC and CEP55 expression in MDA-MB-231 cells upon transfection with siRNA against *MYC* and *ETS1* (10 nM), with and without EGF (100 ng/ml) stimulation as indicated time points. COX-IV served as a loading control.

G  Immunoblots analysis showing CEP55 and MYC levels following 4-hydroxytamoxifen (4OHT) (0.5 μM) induction in MCF10A *MYCER* cells cultured in 0.1% fetal bovine serum contained media at indicated time points. COX-IV served as a loading control.

H  Analysis of Pearson correlation coefficients (PCC) of CEP55 versus MYC levels with $R = 0.9982$, $P = 0.0016$. Protein band intensities as shown in (G) were measured using ImageJ software.

I  Analysis of Pearson correlation coefficients (PCC) of *CEP55* versus *MYC* transcriptional activity in breast cancer TCGA tumor data with $R = 0.4157$, $P < 0.0001$.

J  Both control and CEP55 knockdown MDA-MB-231 cells were exposed with different concentrations of PLK1 (BI2536) alone (i) or in combination with AZD6244 (1 μM) (ii–iv), and cell viability was determined after 6 days. The dose–response curve was generated by calculating cell viability relative to untreated control and plotted against drug concentration. Graph represents the mean ± SEM of three independent experiments.

K  Percentage of sub-G1 population identified using propidium iodide staining and quantified by FACS following single and combination treatment with AZD6244 (1 μM) and BI2536 (2.5 nM) inhibitors after 96 h. Graph represents the mean ± SEM of two independent experiments. *$P \leq 0.05$; ****$P \leq 0.0001$.

L  Immunoblots analysis of both control and CEP55 knockdown MDA-MB-231 cells treated with single and combination treatments after 96 h and cleaved PARP, caspase-3 along with MYC, ERK1/2, and CEP55 were determined. COX-IV served as a loading control.

Source data are available online for this figure.

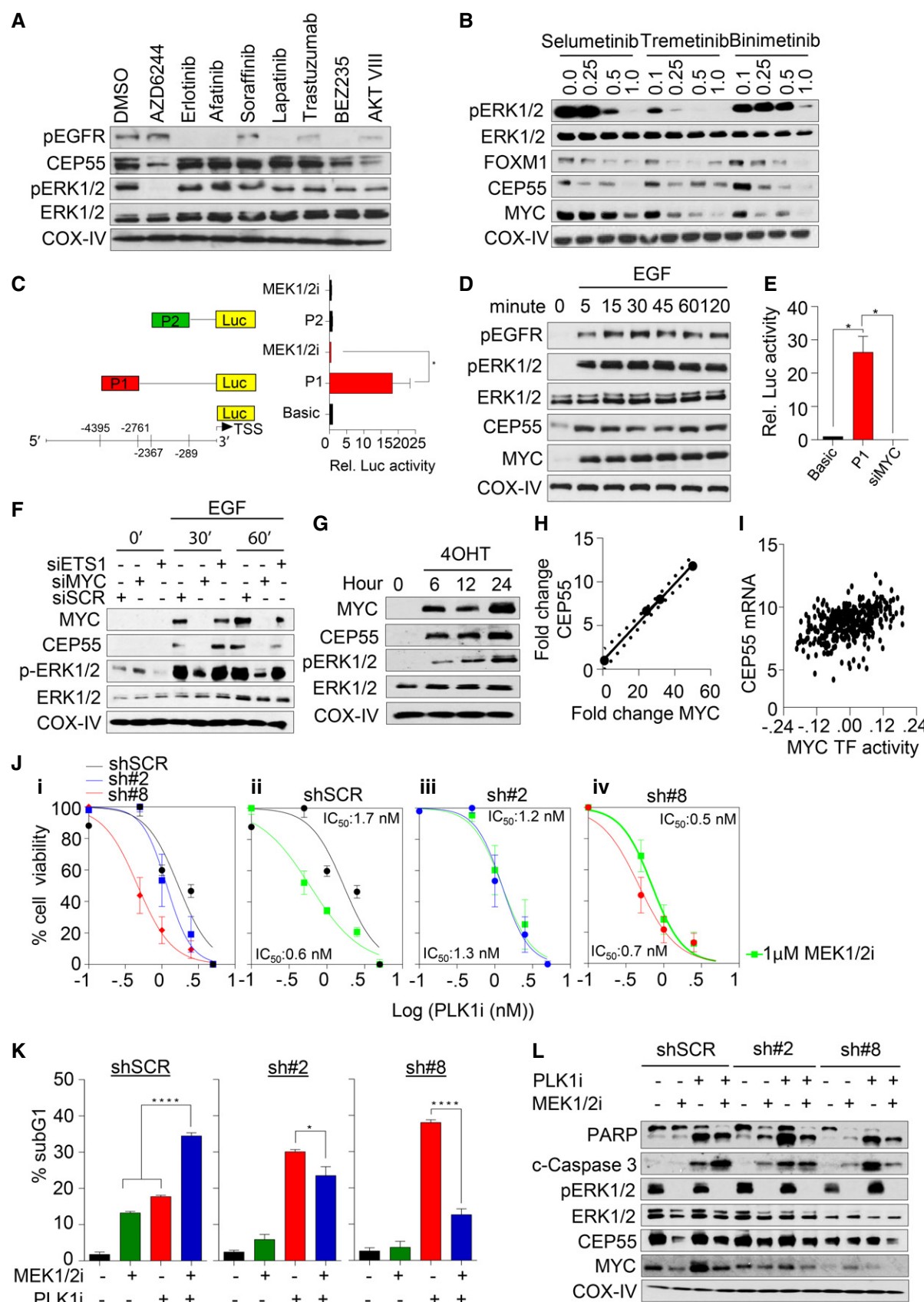

Figure 5.

knockdown cells to anti-mitotic agents, we tested whether CEP55 overexpressing breast cancer cells can be targeted by combination of a selective MEK1/2 inhibitor (AZD6244) and the anti-mitotic drug (PLK1 inhibitor, BI2535). We found that both MDA-MB-231 and Hs578T cells expressing high levels of endogenous CEP55 showed profound sensitivity to the combination treatment (Fig 5J ii, Appendix Fig S8Bii), concomitant with increased apoptosis (subG1, cleaved PARP and caspase-3) (Fig 5K and L, Appendix Fig S8C and D); however, CEP55-knockdown MDA-MB-231 and Hs578T cells were already sensitive to PLK1 inhibition (Fig 5Ji, Appendix Fig 8A and Bi), and did not show increase sensitivity upon MEK1/2 inhibition (Fig 5J iii–iv, Appendix Fig S8Biii). To our surprise, cells with CEP55-loss tend to exhibit reduced cell death phenotype to this combination treatment (Fig 5K, Appendix Fig S8C). Notably, the sensitivity to combination treatment was restored after transient expression of shRNA-resistant CEP55 in MDA-MB-231 (sh#8) cells (Appendix Fig S8E). Moreover, we found a marginal increase in apoptosis upon combination treatment in CEP55-overexpressing MCF10A cells compared to empty vector-transfected cells (Appendix Fig S8F). However, this increase is not in par with cancer cell lines expressing high endogenous CEP55 levels.

Having established that the combination treatment targeted CEP55-dependent cells, we treated a panel of 21 breast cancer lines with these agents and found mixed responses, but basal-like, triple-negative (TNBC) lines specifically responded to the combination treatment (Fig 6A). Additionally, near-normal MCF10A and D492 lines exhibited non-responsive effect following combination treatment, consistent with the idea that CEP55 activity is important in certain cancers but not in normal cells. The combined treatment synergistically induced apoptosis in TNBC lines, reflected by increased subG1 population concomitant with induction of cleaved caspase-3 and PARP (Fig 6B, Appendix Fig S8G, Appendix Table S1), and completely eradicated colony formation in all TNBC lines tested, but not in ER$^+$ (MCF7) or HER$^+$ (SKBR3) lines (Fig 6B, Appendix Fig S8G and H). Moreover, using time-lapse microscopy we found that the combined MEK1/2-PLK1 inhibited cells significantly died in mitosis compared to individual drug-treated cells which either underwent mitotic slippage or divided normally (Fig 6C). In addition, similar to knockdown experiments (Fig 4E), we found that combination treatment markedly reduced CEP55 level and accelerated the G2/M entry

due to premature CDK1/cyclin B activation (evident by phospho-MEK$^{T286}$, accumulation of cyclin B1, and dephosphorylation of CDK1$^{Y15}$) as early as 9 h, causing cell death at 12 h in both MDA-MB-231 and SUM159PT cells (Fig 6D) concomitant with reduced expression of anti-apoptotic proteins (MCL1, BCL2, and survivin). Collectively, these data suggest that CEP55 is a determinant of combined MEK1/2-PLK1i sensitivity and raise the possibility that the combination treatment could have efficacy in CEP55-dependent, triple-negative breast tumors.

### Combined MEK1/2-PLK1 inhibition impedes aggressive tumor formation *in vivo*

Based on the persuasive *in vitro* data, we investigated MEK1/2-PLK1 combination therapy in syngeneic and xenograft models of breast cancer. Immunocompetent BALB/c mice were inoculated with high CEP55-expressing aggressive syngeneic cell line 4T1.2 (mammary fat pad injections). Treatments were initiated when tumors reached 25 mm$^2$: 12.5 mg/kg BI6727 [similar activity as BI2536 and under active clinical investigation (Maire *et al*, 2013; Zhang *et al*, 2016b)], thrice weekly, and 12.5 mg/kg AZD6244 BID for 2 weeks. As observed *in vitro* (Appendix Fig S8G), tumors exhibited modest responses to individual agents, but pronounced outgrowth inhibition in the combination arm (Fig 7A), without noticeable toxicity on their body weight (data not shown). Immunoblot analysis of tumor lysates showed reduced CEP55 and enhanced cleaved caspase-3 (Fig 7B). Next, we pretreated mice with combined inhibitors for 4 days before grafting 4T1.2 cells, followed by a further 10 days of treatment. While mice treated with vehicle or single agents died within 30 days, combination therapy prolonged survival (median 55 days; Fig 7C and Appendix Fig S9A), without noticeable toxicity in any of the treatment groups (data not shown). Furthermore, NOD/SCID fat pad xenografts of MDA-MB-231 and its metastatic variant HM-LNm5 (both aneuploid lines) exhibited profound tumor regression and longer survival with combination therapy (Fig 7D and E, Appendix Fig S9B). In summary, these data show that blocking MEK1/2/PLK1 substantially impacts tumor growth in multiple models of basal-like, triple-negative breast cancer and represents a novel candidate strategy for treating breast tumors that exhibit high CEP55 expression.

---

**Figure 6.  Combined MEK1/2-PLK1 inhibition specially kills basal-like breast cancer lines.**

A   Heat map showing relative cell viability of a panel of human breast cancer lines treated single or in combination with AZD6244 (1 μM) and BI2536 (2.5 nM) inhibitors and cell viability was determined after 6 days. DMSO-treated control was used to calculate percentage of cells affected by individual or combination treatment. NN: near normal. Heat map represents data derived from three independent experiments.

B   Immunoblot analysis was performed on MDA-MB-157 and SUM159PT cells treated with single or in combination with AZD6244 and BI2536 inhibitors after 96 h and cleaved PARP and caspase-3 were determined along with CEP55, MYC, and both phosphorylated and total ERK1/2. COX-IV as a loading control (left panels). Percentage of sub-G1 population identified using propidium iodide staining and quantified by FACS following single and combination treatment with AZD6244 and BI2536 inhibitors after 96 h. Graph represents the mean ± SEM of two independent experiments (middle panels). Representative images of colony-forming capacity at 14 days determined using crystal violet staining in cells treated with single and combination inhibitors (middle panels). Combination index (CI) values calculated through CompuSyn software (right panels) where 1 indicative of additive (no interaction), > 1 indicative of antagonistic, and < 1 indicative of synergistic. ****$P \leq 0.0001$.

C   Percentage of mitotic outcomes determined using time-lapse microscopy as described in Fig 4A of MDA-MB-231 cells treated with single or in combination drugs of BI2536 (2.5 nM) and AZD6244 (1 μM). Graph represents the mean ± SEM of two independent experiments. For each experiments, $n$ = 50 cells were counted per condition. ****$P \leq 0.0001$.

D   Immunoblot analysis was performed to determine the expression and activity of mitotic and apoptotic regulators upon single and combination treatment with AZD6244 and BI2536 inhibitors as indicated time points in both MDA-MB-231 and SUM159PT cells. Levels of phospho-MEK$^{T286}$ and dephosphorylation of phospho-CDK1$^{Y15}$ served as markers of CDK1 activation/mitotic entry. COX-IV served as a loading control.

Source data are available online for this figure.

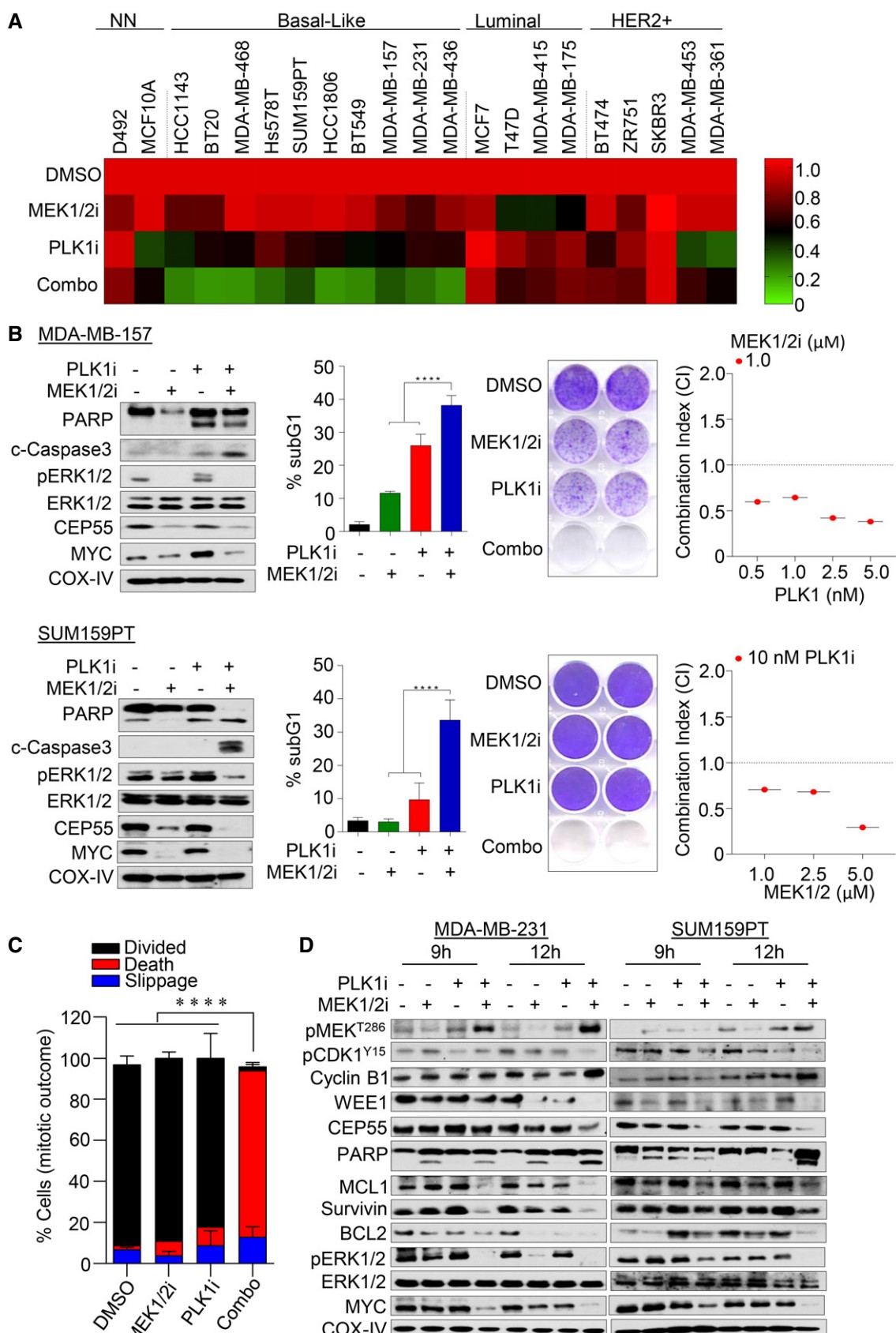

**Figure 6.**

**Figure 7.  Co-blockade of MEK1/2 and PLK inhibits basal-like tumor outgrowth *in vivo*.**

A    Left, 6-week-old female BALB/c cohorts of mice were injected in the 4[th] inguinal mammary fat pad with the Cep55-overexpressing mammary carcinoma cell line 4T1.2. Tumor size (area, mm$^2$) was measured using a digital calliper and mean tumor size of each cohort is presented. Mice were treated with vehicle, AZD6244 (12.5 mg/kg BID), BI6727 (12.5 mg/kg thrice weekly), or combined AZD6244 and BI2536 treatment. Right, representative excised tumor images are shown. Graph represents the mean tumor area $\pm$ SEM from six mice/group. ****$P \le 0.0001$.

B    Immunoblot analysis was performed on tumor lysates to determine target inhibition following single and combined treatments as indicated in panel (A). Cleaved caspase-3 was probed to determine apoptosis. GAPDH served as a loading control.

C    Six-week-old female BALB/c cohorts of mice were pretreated with single and combination BI6727 and AZD6244 inhibitors as indicated in panel (A) for 4 days, followed by 4T1.2 cells were injected in the 4[th] inguinal mammary fat pad. Survival of the mice was then monitored over the indicated period of time and the statistical significance of data was analyzed by log-rank test ($P = 0.0005$); $n = 6$ mice/group.

D, E    Growth rate (mean tumor size, area, mm$^2$) (D) of MDA-MB-231 xenografts in BALB/c nude mice treated with vehicle, AZD6244, BI6727, or combined AZD6244 and BI6727 treatment as indicated in panel (A), and (E) survival of mice was monitored over the indicated period of time and the statistical significance of data was analyzed by log-rank test ($P < 0.0001$); $n = 6$ mice/group. **$P \le 0.01$; ****$P \le 0.0001$. Mean $\pm$ SEM.

F    Schematic overview of MAPK pathway in controlling CEP55 level through c-MYC. Co-blockade of PLK1 (BI6727), a mitotic blocker with MEK1/2 (AZD6244), represents a potential therapeutic strategy to treat MYC-CEP55-dependent aggressive basal-like, triple-negative breast cancers.

## Discussion

Elevated levels of mitotic proteins contribute to early cellular transformation and tumorigenesis (Kops *et al*, 2005). Mitotic genes are rarely mutated in cancer (Cahill *et al*, 1999; Hernando *et al*, 2001), but rather affected by CNAs; for example amplification of *PLK1* (Strebhardt & Ullrich, 2006; Degenhardt & Lampkin, 2010), Aurora-A (Zhou *et al*, 1998b; Wang *et al*, 2006), survivin (Altieri, 2003), cyclin B (Malumbres & Barbacid, 2009), and *NEK2* (Hayward *et al*, 2004). Despite numerous reports demonstrating CEP55 overexpression in cancer (Jeffery *et al*, 2015), the precise mechanistic link with genomic instability has not been previously investigated and therefore is currently not well defined.

Here, we provide the first detailed description of CEP55's role in promoting genomic instability in breast cancer. We found that CEP55 is highly expressed in breast cancer with poor clinical outcomes. Inoda *et al* (2009) described CEP55 as a cancer testis antigen and postulated that CEP55-derived peptides could be used as vaccine-based therapy in breast and colorectal cancers. CEP55 has been shown to regulate cell proliferation, migration, and invasion in multiple cancer cell line models (Jeffery *et al*, 2015). Consistent with this, we found that reduced/downregulated CEP55 levels impaired cell proliferation, clonogenic potential, migration, and invasion *in vitro*. Conversely, exogenous overexpression enhanced colony-forming ability in 2D and 3D cultures, with elevated proliferation. The fact that multiple different tumorigenic behaviors were affected by perturbing CEP55 expression suggests that, like other mitotic proteins [e.g., Aurora-A (Fu *et al*, 2007), Eg5 (Yu & Feng, 2010), PLK1 (Ito *et al*, 2004; Weichert *et al*, 2005), NEK2 (Zhong *et al*, 2014)], it could regulate multiple aspects of breast cancer development.

A century ago, Theodor Boveri observed aneuploidy in sea urchin embryos that divided abnormally, impacting cell growth and survival (Holland & Cleveland, 2009). Aneuploidy and genomic instability are now considered independent but significantly overlapping events in tumorigenesis (Sieber *et al*, 2003; Potapova *et al*, 2013). *CEP55* is a component of a 70-gene signature associated with chromosomal instability in several human cancers (Carter *et al*, 2006). Consistent with this, we found that even partial silencing of *CEP55* in aggressive basal-like breast cancer cell lines eliminated their aneuploid subpopulations as a response to mitotic cell death, while allowing near-diploid cells to survive in culture. This experiment mimicked the phenotype that arises from knockdown of *MPS1/*

*TTK*, a core component of SAC (Daniel *et al*, 2011). Kimura *et al* (2013) demonstrated that silencing of centrosomal proteins triggered aberrant mitoses, prompting cell death via a SAC-dependent mechanism, regardless of p53 status. Our data demonstrated a role of CEP55 in aneuploidy tolerance, since knockdown of CEP55 in breast cancer lines resulted in elimination of aneuploid population. Moreover, CEP55-knockdown cells were unable to tolerate prolonged mitotic arrest induced by mitotic poisons and instead, died in mitosis. Notably, CEP55-knockdown sensitized cells to anti-mitotic drugs through premature CDK1/cyclin B activation, which is crucial to cell death pathway activation (Shi *et al*, 1994; Zhou *et al*, 1998a).

Resistance to anti-mitotic drugs like taxol (Chan *et al*, 2012) represents a significant clinical problem in the management of breast cancer patients. Although most but not all tumors show some degree of chemotherapy response, tumor cells often bypass mitotic arrest and prematurely exit mitosis, a phenomenon known as mitotic slippage. Mitotic slippage has been shown to be a *bona fide* mechanism of resistance to taxol and other mitotic agents (Chan *et al*, 2012; Burgess *et al*, 2014). Consistent with this, CEP55 overexpression provided a survival advantage in the context of mitotic insult *in vitro* by enabling cells to evade apoptosis; and conversely, knockdown of CEP55 sensitized cells to anti-mitotic agents. We found that *CEP55* mRNA expression stratified outcome among patients treated with various chemotherapy drugs. Thus, apart from providing mechanistic insights, these data have potential clinical relevance. Future studies should address the utility of *CEP55* expression and other CIN70 genes as a predictive indicator of response to taxols, and possibly other anti-mitotic drugs.

A major obstacle in the clinic is the relatively shallow armamentarium of systemic agents for treating triple-negative breast cancer (TNBC). Identifying targetable pathways that these tumors are dependent on for survival is a high priority in the field. Multiple deregulated pathways have been identified but targeting these with individual agents has produced disappointing results in preclinical and clinical trials (Kalimutho *et al*, 2015; Wali *et al*, 2017). For example, MEK1/2 blockade activates RTK-mediated resistance within hours of treatment, while co-blockade of RTKs and MEK1/2 (sorafenib+AZD6244) was more effective in killing TNBC tumors (Duncan *et al*, 2012). High rate of proliferation and genomic instability are features common to many TNBCs, and while cell cycle-targeting agents have generally failed in previous trials, accurate analysis of efficacy was potentially confounded by trial design issues. More recently, the possibility of targeting mitosis and mitotic

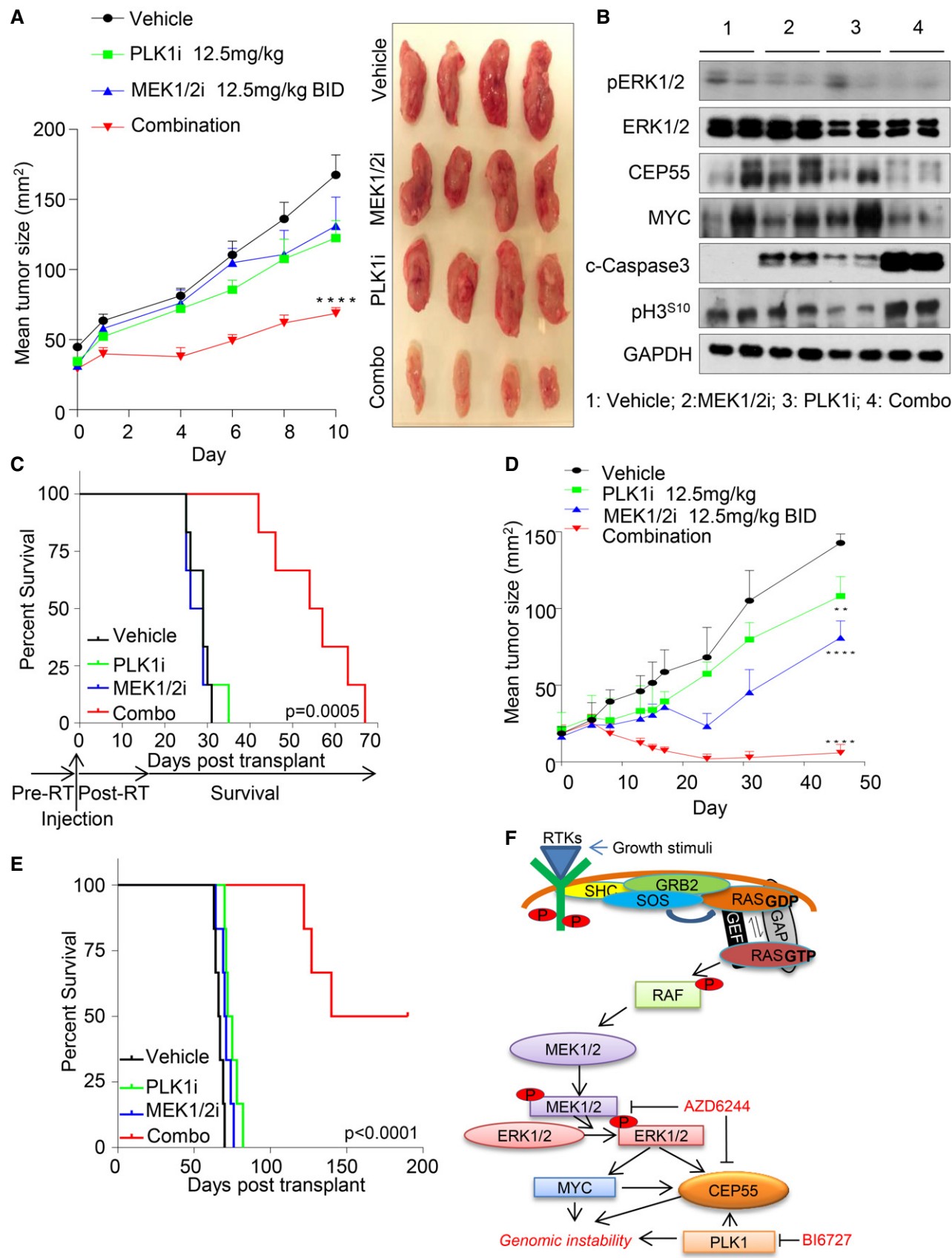

Figure 7.

exit has received more attention. Inhibitors of cyclin-dependent kinase (CDK) 1, CDK 4/6, auroras, polo kinases, or spindle kinesins are being actively investigated for better therapeutic efficacy in appropriately selected patient groups (Manchado *et al*, 2012; Dominguez-Brauer *et al*, 2015; Lim *et al*, 2016).

We found that CEP55 is a downstream effector of MAPK signaling through MYC oncogene (Fig 7F), which has a central role in transformation, tumorigenesis, and genomic instability, and MYC is deregulated in many cancers (Sears *et al*, 2000; Prochownik, 2008). In breast cancer, deregulation of MYC is most frequent in basal-like tumors and is associated with resistance to adjuvant chemotherapy (Xu *et al*, 2010); hence, targeting its dependent pathways could be a good therapeutic strategy. Combining inhibitors of MEK1/2 (targets the MYC-CEP55 axis) and PLK1 (mitosis) inhibited the outgrowth of aggressive basal-like syngeneic and human breast cancer xenografts. The effects of similar combination strategy have also been reported in two other studies: MEK1/2i with docetaxel (a microtubule poison) reduced mammary tumor growth *in vivo* (Yacoub *et al*, 2006), and a MEK1/2-PLK1 inhibition caused regression of *NRAS* mutant melanoma xenografts (Posch *et al*, 2015).

Unlike other spindle checkpoint proteins [e.g., Mps1 (Daniel *et al*, 2011), Bub1 (Ricke *et al*, 2011), and Mad2 (Kops *et al*, 2004)], centrosomal proteins have not been previously implicated in aneuploid cell survival. This study is the first to demonstrate that CEP55 can confer a survival benefit by evading apoptosis during perturbed mitosis. Targeting mitotic proteins has long been known to be an effective therapeutic strategy in highly proliferating tumor cells (Chan *et al*, 2012). However, strategies to target mitotic proteins have failed in the clinic due to various reasons, including unselected patient cohorts, lack of companion biomarkers, toxicity, and drug resistance (Chan *et al*, 2012). To address these hindrances, we provide strong preclinical evidence in favor of a novel therapeutic approach targeting aneuploidy and genomic instability, which are fundamental and recurrent features of TNBCs. Since various PLK1 inhibitors are already being assessed individually in clinical trials and MEK1/2 inhibitor trametinib is recently approved by FDA for treating patients with metastatic melanoma, our data suggest that a combination strategy could have better efficacy in basal-like, TNBCs that exhibit highly aberrant copy-number profile.

# Materials and Methods

## Reagents

RO-3306, nocodazole, BI2536, BI6727, AZD6244, erlotinib, afatinib, sorafenib, lapatinib, trastuzumab, BEZ235, and AKTVIII were purchased from Selleck Chemicals LCC. Small interfering RNAs (siRNAs) were from Shanghai Gene Pharma. Lipofectamine RNAiMAX was purchased from Life Technologies. CellTiter 96® AQueous One Solution Cell Proliferation Assay and Dual-Glo® Luciferase Assay were purchased from Promega Corporation.

## Cell culture

The breast cancer cell lines used in this study were purchased from the American Type Culture Collection (ATCC), otherwise stated in acknowledgment, cultured and maintained as per ATCC

recommendations and as described previously (Anderson *et al*, 2017). All the cell lines were regularly tested for mycoplasma infection and authenticated using short tandem repeat (STR) profiling by scientific services at QIMR Berghofer Medical Research Institute.

## Constructs and mutagenesis

Generation of Flag-CEP55 and constitutive shRNA plasmids used in this study were reported previously by us (Fabbro *et al*, 2005). The shRNA-resistant mutant construct (Indicated as Rescue in figures) was created by site-directed mutagenesis using QuikChange XL-II kit (Stratagene) according to the manufacturer's instructions using primer sequence of 5′-cagatatagtactaccacattgctAgaGcaActAgaagagacaacgagagaaggagaa-3′. Mutant construct was checked by sequencing as shown in Fig EV2D. CEP55 promoter luciferase constructs were kindly provided by Prof. Pin Ouyang (Chang Gung University).

## Generation of CEP55-overexpressing MCF10A lines

MCF10A cells were reverse-transfected either with Flag-CEP55 expression construct using 1.0 µg of plasmid described previously (Fabbro *et al*, 2005), and the stably CEP55-overexpressing lines were selected using 0.5 µg/ml puromycin.

## Generation of CEP55-shRNA lines

Doxycycline-inducible shCEP55 MDA-MB-231-luc cells were established using the lentiviral Invitrogen BLOCK-iT™ Pol II miR RNAi Expression vector kit (Life Technologies). Two different CEP55 shRNAs with its respective scrambled shRNA sequences were used as shown below. Knockdown cells were selected using 50 µg/ml zeocin and 5 µg/ml Blasticidin. In addition, constitutively CEP55-knockdown in Hs578T cells was achieved using previously reported plasmids (Fabbro *et al*, 2005).

shRNA#2
5′-TGCTGTAAGCATTCTTCTCCTTCTCAGTTTTGGCCACTGACTGACTGAGAAGGAAGAAT CTTA-3′

shRNA#8
5′-TGCTGTCTTCCAGCTGTTCAAGCAATGTTTTGGCCACTGACTGACATTGCTTGCAGCTGGAAGA-3′

## Small interfering RNAs (siRNAs) and cell viability

Breast cancer cell lines were plated in 96-well plates at 5,000–8,000 cells/well followed by reverse transfection using 5–10 nM of siRNAs (Appendix Table S2) for 6 days, and cell viability was measured by CellTiter 96 AQueous One Solution Cell Proliferation Assay Kit as described previously (Al-Ejeh *et al*, 2014).

## Reverse transcriptase–quantitative PCR

RNA was extracted using RNEasy Mini Kit (Qiagen, Venlo, Limburg, the Netherlands) and cDNA synthesized using the SuperScript III First-Strand Synthesis System (Life Technologies) according to manufacturer's instructions. RT–qPCR was performed on a Light-Cycler 480 (Roche, Basel, Switzerland) using SYBR Green (Roche) and normalized against β-actin and HPRT1 as an internal control (Appendix Table S3).

## Immunoblotting

Immunoblotting was performed as described previously (Van Schaeybroeck *et al*, 2014) with indicated antibodies (Appendix Table S4). The Super Signal chemiluminescent ECL-plus (Amersham) was used for detection.

## Flow cytometry

Cell cycle perturbations and the subG1 apoptotic fractions were determined using flow cytometry analysis of cells stained with propidium iodide and analyzed using ModFit LT 4.0 software as described previously (Kalimutho *et al*, 2011).

## *In vitro* cell migration and invasion assays

Cell migration and invasion (with BD growth factor reduced Matrigel) rates were assessed using XCELLigence system (Roche Applied Sciences) as described previously (Dunne *et al*, 2014).

## Colony formation assays

Five hundred to one thousand cells were seeded on 12-well plates and incubated for additional 14 days to determine colony viability. The colonies were fixed with 0.05% crystal violet for 30 min, washed, and quantified for crystal violet intensity after destaining using Sorenson's buffer (0.1 M sodium citrate in 50% Ethanol, pH 4.2) at 590 nM absorbance using PowerWave HT Microplate Spectrophotometer (BioTeK, USA).

## Ingenuity pathway analysis

Ingenuity pathway analysis was performed using the Ingenuity Pathway Analysis® (IPA) software (Ingenuity Systems®, Redwood City, CA).

## 3D-acini culture and imaging

The 3D acini using MCF10A cells were performed using well-established techniques as described previously (Debnath *et al*, 2003). Single-plane phase-contrast and Z-stacked immunofluorescence images were acquired through Zeiss LSM 780 confocal microscope-ZMBH.

## SNP arrays

SNP array analysis (Omni2.5M BeadChips, Illumina) was performed according to the manufacturer's protocol using 200 ng of DNA. Arrays were scanned using an iScan (Illumina). SNP arrays data were processed using the Genotyping module (v1.9.4) in GenomeStudio v2011.1 (Illumina, San Diego CA) to calculate B-allele frequencies (BAF) and logR ratios. The GAP tool (Popova *et al*, 2009) was used to estimate copy-number change. Copy-number segments were classified as: amplified (copy number 6–8); gained (copy number 3–5); lost (copy number 0–1); or copy-neutral LOH (copy number 2 with major allele contribution of 0 or 1). Genes within regions of copy-number change were annotated using ENSEMBL v70.

## Live-cell imaging

Live-cell imaging was performed on an Olympus IX81 microscope using excellence rt v2.0 software. Images were analyzed using analySIS LS Research, version 2.2 (Applied Precision) as described previously (Jeffery *et al*, 2013).

## CDK1 biosensor FLIM-FRET imaging

Cells were transfected with pcDNA CDK1 biosensor (Gavet & Pines, 2010) using Lipofectamine 3000. Imaging was performed as described previously (Vennin *et al*, 2017). Briefly, FLIM-FRET signal was acquired using a 25× 0.95 NA water objective on an inverted Leica DMI 6000 SP8 confocal microscope. Excitation source was a Ti: sapphire femtosecond laser cavity (Coherent Chameleon Ultra II), operating at 80 MHz. mCerulean was excited at 840 nm, and the signal was detected using a 490/40 nm filter. FLIM data were recorded using a Picoharp 300 TCSPC system (Picoquant). Images of 512 × 512 pixels were acquired with a line rate of 600 Hz. The pixel dwell time was 5 μs, and images were integrated until 600 photons per pixel were acquired. mCerulean lifetimes were analyzed by drawing ROIs around subcellular areas and recording the lifetime (τ) of the single exponential function fit to the fluorescence decay data. Lifetime maps were generated with intensity thresholds set to the average background pixel value for each recording. The raw data were smoothed, and a standard rainbow color look-up table (LUT) was applied, with lifetimes of 1 ns (blue) to 3 ns (red) for mCerulean. Areas where no lifetime measurement above the background noise could be achieved are shown in black in the lifetime map.

## METABRIC and TCGA datasets and copy-number alterations (CNAs)

Clinical annotations, processed values of gene expression, and CNA profiles were downloaded from METABRIC (Curtis *et al*, 2012) (Discovery $n = 995$) and TCGA (Cancer Genome Atlas N, 2012) ($n = 492$). To calculate CNA for both datasets, we used the available copy-number value overlapping genes in a given tumor sample. We computed an average across ~20,000 genes to obtain average CNA for each tumor (herein defined as average copy number of genome/tumor). These average values were used to estimate Pearson correlation between CEP55 mRNA expression levels and average CNA across tumors and are the values used in Fig 2F and H, Appendix Fig S4P. MYC transcription factor activity was inferred from TCGA tumor samples using their protein expression profiles and a previously described trained affinity regression model (Osmanbeyoglu *et al*, 2017). Proliferation-adjusted CEP55 expression was calculated by dividing the CEP55 mRNA level by the MKI67 or PCNA mRNA level, all expressed in RPKM and RSEM-normalized.

## *In vivo* xenografts

All experiments were in accordance with the guidelines of the QIMR Berghofer Medical Research Institute Animal Ethics Committee. 5–6 weeks of female NOD/SCID or BALB/C mice were used in this study. All mice were housed in standard condition with a 12-h light/dark cycle and free access to food and water. For mammary

## The paper explained

### Problem

The link between aneuploidy, a phenomenon of abnormal chromosome constitution, and cancer has been recognized of fundamental importance and remains poorly understood. A number of cellular mechanisms that lead to aneuploidy have now been well characterized including defects in faithful segregation of chromosomes during mitosis. Exploring how mitosis-dependent genes facilitate these processes in tumor evolution has the potential to uncover a major mechanism of tumor cell heterogeneity and treatment resistance, thus offering potential aneuploidy-specific targets in the cancer therapeutics armamentarium. From a clinical perspective, aneuploid cancers generally have poor clinical outcomes, due to its association with metastasis and therapeutic resistance.

### Results

We identified high levels of CEP55, a cytokinesis regulator, promote survival of aneuploid breast cancer cells. Knockdown of CEP55 sensitizes them to anti-mitotic agents through premature CDK1/cyclin B activation and CDK1 caspase-dependent mitotic cell death. Furthermore, we showed that CEP55 is a downstream effector of the MAPK-MEK1/2-MYC axis. We found that inhibition of two well studied oncogenic pathways (MEK1/2 and PLK1 axes) in breast cancer significantly reduced outgrowth of basal-like syngeneic and human breast tumors xenografts in preclinical models. To conclude, this study identified a novel role of oncogenic CEP55 in dictating cell fate during perturbed mitosis. Forced mitotic cell death by blocking MEK1/2-PLK1 represents a potential therapeutic strategy for MYC-CEP55-dependent basal-like, triple-negative breast cancers.

### Impact

Targeting aneuploidy and genomic instability has the potential to significantly impact clinical practice in the treatment of several cancers. Although about 90% of all solid human tumors contain numerical chromosome aberrations, no specific targeted therapies are available to treat these cancers. This study provides experimental evidence that targeting a specific cellular phenotype driven by potent oncogenes such as CEP55 indeed can be utilized for killing aggressive breast cancers with aberrant DNA contents. This study sheds light not only on the mechanism contributing to aneuploidy and cancer pathogenesis, but will also improve our understanding on therapeutic options in advanced stage diseases.

fat pad injections, $2.5 \times 10^6$ human MDA-MB-231 including the CEP55-depleted lines and $1.0 \times 10^6$ MDA-MB-231-HM_LNm5 cells were prepared in 50% Matrigel (BD, Biosciences, Bedford, USA)/PBS and injected into the right $4^{th}$ inguinal mammary fat pad of 6-week-old BALB/C nude mice. For 4T1.2 tumor cells injection in BALB/C mice, $1.0 \times 10^6$ were prepared in PBS. Tumor growth was measured by thrice weekly by calliper measurements. To calculate tumor area the following formula was used: tumor area = B*S where B = largest tumor measurement and S = the smallest, based on two-dimensional caliper measurements.

### Statistical analysis

Student's *t*-tests, one-way or two-way ANOVA with post hoc Bonferroni, log-rank test were performed where appropriate using GraphPad Prism v6.0 and the *P*-values indicative of ns $P > 0.05$; $*P \leq 0.05$; $**P \leq 0.01$, $***P \leq 0.001$, and $****P \leq 0.0001$. Exact individual *P*-values and the statistical test used for each analysis are shown in Appendix Table S5. Mean ± SEM: Mean and standard error of mean are used to describe the variability within the sample in our analysis.

## Data availability

The dataset produced in this study is available in the following database:

- SNP array: Gene Expression Omnibus GSE117058 (https://www.ncbi.nlm.nih.gov/geo/query/acc.cgi?acc = GSE117058)

**Expanded View** for this article is available online.

## Acknowledgements

MK is supported by Cancer Council Queensland (CCQ) project grant [ID 1087363]. KK laboratory is supported by National Health & Medical Research Council (NH&MRC) Program Grant [ID 1017028]. KN is supported by Keith Boden Fellowship. DS is supported by Griffith University International and Postgraduate Research Scholarship. PT is supported by funding from NHMRC, Cancer Council NSW, Cancer Australia, Tour de Cure grants, Cancer Institute NSW, ARC Future, Lens Ainsworth, and CV is supported by Sydney Catalyst Scholarships. We thank the members of the Khanna laboratory for helpful discussions and technical assistance, Amanda Bain for thoroughly editing this manuscript, Stephen Miles for maintaining cell lines, Masroor Shariff for developing CEP55-overexpressing MCF10A cells, QIMR Berghofer Flow Cytometry and Animal facility staffs, Nigel Waterhouse and Tam Hong Nguyen from ACRF Imaging Centre for microscopic assistance, and Paul Collins for STR profiling and MYCoplasma testing, Prof. Jean Gautier for providing MCF10A-*MYCER* cells, Prof. Pin Ouyang for providing CEP55-luciferase constructs, Prof. Thorarinn Gudjonsson for providing D492 cells, and Dr Cameron Johnstone for providing MDA-MB-231-HM-LNM5 cell line.

## Author contributions

Conceptualization, MK and KKK; investigation and data analysis, MK, DS, JS, KN, CV, PT, DM, NW, PR, DN, and WCF; bioinformatics, SS, PHGD; writing–original draft, MK, and KKK; writing–review and editing, all authors; funding acquisition, MK, and KKK; resources, JMS, SRL, JAL, KJS, and BG. All authors read and approved the final manuscript.

## Conflict of interest

The authors declare that they have no conflict of interest.

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
