## [Review Process File · EMBO Molecular Medicine]

CEP55 is a determinant of cell fate during perturbed mitosis in breast cancer

Murugan Kalimutho, Debottam Sinha, Jessie Jeffery, Katia Nones, Sriganesh Srihari, Winnie C. Fernando, Pascal H.G. Duijf, Claire Vennin, Prahlad Ranninga, Devathri Nanayakkara, Deepak Mittal, Jodi M. Saunus, Sunil R. Lakhani, J. Alejandro López, Kevin J. Spring, Paul Timpson, Brian Gabrielli, Nicola Waddell, Kum Kum Khanna

Review timeline:

Submission date:	09 October 2017
Editorial Decision:	06 December 2017
Revision received:	27 April 2018
Editorial Decision:	15 June 2018
Revision received:	15 July 2018
Accepted:	18 July 2018

Editor: Céline Carret

Transaction Report:

1st Editorial Decision

06 December 2017

Thank you for the submission of your manuscript to EMBO Molecular Medicine and for your patience while the study was being peer-reviewed. We have now heard back from the two referees whom we asked to evaluate your manuscript.

Although the referees find the study to be of potential interest, they also raise a number of significant concerns about the limited mechanistic insights provided (ref.1) and lack of rescue experiments (ref.2). Both referees make detailed comments and we would like to encourage you to address them all to the best of your availabilities. Please note that during our cross-commenting exercise, technical concerns emerged and the contribution of p53 in this setting should be better determined.

We would welcome the submission of a revised version within three months for further consideration and would like to encourage you to address all the criticisms raised as suggested to improve conclusiveness and clarity. Please note that EMBO Molecular Medicine strongly supports a single round of revision and that, as acceptance or rejection of the manuscript will depend on another round of review, your responses should be as complete as possible.

Please read below for **important** editorial formatting and consult our author's guidelines for proper formatting of your revised article for EMBO Molecular Medicine.

I look forward to receiving your revised manuscript.

***** Reviewer's comments *****

Referee #1 (Comments on Novelty/Model System for Author):

The technical quality is medium because they could have made more effort to deepen molecular signalling that control CEP55 or linking CEP55 to molecular pathways driving breast cancer subtypes.

Novelty is low because there are previous papers (some of them by the authors themselves) describing CEP55 roles in aneuploidy.

Referee #1 (Remarks for Author):

The study shows that loss of CEP55 could sensitize breast cancer cells to anti-mitotic agents, suggesting further putative therapeutic avenues to treat breast cancer.

Although the findings of the manuscript are interesting as well as their potential in breast cancer therapy, however the weak of the manuscript relies on the loss of mechanistic novel insights. Indeed, in reviewer's eye the manuscript does not provide substantial novel insights about the molecular mechanisms involving CEP55, p53, Plk1 and MAP-kinase cascade.

The above mentioned has been the main concern that reviewer could raise about the manuscript. The minor concerns are about the description of experimental procedures: although the technical quality of the manuscript is good, however sometimes it is very difficult to understand the experimental design and the reasons of some experimental choices.

In particular:

1- it is true for the first part of the manuscript running from the row 109 to 275.

Authors should explain what are their main objectives and how they decide to pursue them. This could make the experimental procedures more understandable to readers.

2- Moreover authors should explain the reason why they use a "leaky" system to induce CEP55-depletion and what is the meaning of the different results they have obtained. (row 148-176)

"The inducible knockdown system used in this study was 'leaky' as evidenced by reduced CEP55 expression in both sh#2 and sh#8 polyclonal lines in the absence of doxycycline compared to scrambled shRNA-transfected cells", however it is not clear the reason why they use this leaky system.

3- Authors should explain why "breast cancer cells partially-depleted of CEP55 showed no significant changes in the number of cells displaying cytokinesis failure. In contrast near complete depletion of CEP55 resulted in a significant increase in multinucleated cells (data not shown)" Are those findings dependent on a pleiotropic effect of CEP55? After complete depletion of CEP55 multinucleated cells generated, are multinucleated cells living and aneuploid?

4- At row 182 authors say that p53 depletion is also necessary in order to make MCF10A-CEP55+, able to form tumor in nude mice, however to make the experimental procedure more clear to readers, authors should explain the reasons of this choice.

5- Authors tested whether the CEP55-overexpressing MCF10A cells had acquired an oncogenic phenotype by measuring their ability to form colonies in non-adherent conditions and form acinar structures in Matrigel™.

However it should be more appropriated refers to colonies in matrigel as mammospheres or spheroids. If cell colonies are generated after divisions of aneuploid cells, they will be, more probably, undifferentiated and unable to generate acini. Indeed what authors show in figure 1K are not acini rather spheroid colonies of cells exhibiting anchorage-independent growth. Moreover authors do not provide an immunofluorescence analysis of luminal markers proving that cell colonies are acini.

6- Authors say that CEP55 depletion both in HeLa (previous published data) and MDA-MB-231 reduced aneuploid cell subpopulation however they also say that CEP55-depletion generated multinucleated cells. How could they explain that result?

7- From row 255 up to the end of the "results" section the description of the experimental procedures and results appear more clear.

Referee #2 (Comments on Novelty/Model System for Author):

Several cell line models were used to study cell growth in vitro and in mice

Referee #2 (Remarks for Author):

In this manuscript, Kalimutho and colleagues described the characterization of the effects of CEP55 expression in breast cancer. By analysing public databases, the authors found that high expression of CEP55 mRNA is associated with poor clinical outcomes in breast cancer. They also found that CEP55 is highly expressed in basal-like breast cancer cell lines. The authors then performed a series of experiments to study the effects of CEP55 downregulation and overexpression on cell growth, genome instability, and sensitivity to anti-mitotic drugs. They also found that inhibitors of MEK1/2 reduced the mRNA and protein expression of CEP55. Evidence including using siRNA against MYC indicated that the MEK1/2i effect may be acting through the transcription factor MYC. The authors then demonstrated that combining a MEK1/2i (AZD6244) and a PLK1i (BI2535) sensitized control cells but not partially CEP55-depleted cells. Finally, mouse xenograft models were also sensitized by MEK1/2i and PLK1i.

CEP55 is highly expressed in many cancers and is associated with chromosomal instability. Understanding its role in breast cancer and exploiting this pathway in cancer therapies are important issues. In general, the results are clear and the experiments are well-designed. However, a number of issues reduce my enthusiasm and I am not convinced by some of the authors' claims. These include the lack of rescue experiments and some converse CEP55-overexpressing experiments. The manuscript attempts to cover a wide range of topics (some thinly), hence some results are rather preliminary in nature and require additional investigation.

Points

(1) The authors mainly used two clones of MDA-MB-231 expressing shRNA against CEP55 (using the leakage of the inducible promoter) to study the effects of CEP55 downregulation. In some experiments, the difference in expression of CEP55 between clone#2 and #8 is not so obvious as that shown in Fig 1D (e.g. Fig S2F). The authors should quantify the percentage of knockdown in these two clones relative to the parental cells (e.g. using Western blots with standard curves). Furthermore, referring these cells as "CEP55-depleted cells" in the manuscript is misleading. Another important information appears to be missing is whether is the level of CEP55 after downregulation with shRNA was comparable to cells with low CEP55 (MCF-10A)?

(2) Conversely, the authors generated MCF-10A clones overexpressing CEP55 (Fig 1H). Is the level of the exogenous CEP55 comparable to the endogenous CEP55 found in other cell lines that normally express CEP55 (such as MDA-MB-231)?

(3) A major shortcoming of this manuscript is the majority of the experiments were performed using the two clones of MDA-MB-231 expressing CEP55 shRNA. No rescue experiments (re-introducing CEP55) were performed.

(4) The authors demonstrated CEP55 downregulation resulted in reduced proliferation, anchorage-independent colony formation, migration and invasion, and increased chromosomal stability. Some experiments were then performed using CEP55-overexpressing cells to consolidate the findings (e.g. overexpression of CEP55 induced chromosomal instability of chromosome 20). Why CEP55 overexpression experiments were not performed for other growth characteristics?

(5) The authors found that partially CEP55-depleted cells entered G2/M faster after incubation with nocodazole or PLK1i (but not in unperturbed cells). The data are intriguing but not very convincing. The authors should use live-cell imaging to measure the time of mitosis carefully. Again, converse experiments using CEP55-overexpressing cells will be useful. The comment of the lack of experiments using CEP55-overexpressing cells also apply to when the authors examine the effects of combined MEK1/2i and PLK1i treatment.

(6) The authors showed that depletion of p53 in MCF10A increased CEP55 expression. However, they are largely preliminary in nature and require additional investigation to reveal potential functional links between p53 and CEP55. For example, the authors should examine the contribution of the increase in CEP55 in the cellular effects of p53 depletion.

(7) The authors believed that the apoptosis during mitosis in partially CEP55-depleted cells involved BCL-XL and BIM. These data are too preliminary for the authors to draw this conclusion. Converse experiments of CEP55 overexpression were also not performed.

(8) The meaning of error bars and the number of samples are often missing in the Figure legends (e.g. 2A, 2D, 2F, 3B, 3C, 3F, 3H etc.). Where indicated, they are not very consistent (e.g. the end of the legends of Fig 1 indicates n=2-3; but this probably is not true for part G). Furthermore, some of the experiments with n=2 need more repeats to justify using error bars).

(9) Fig 7F: in the model, the authors indicate a role of PLK1 in controlling CEP55 functions. However, conclusions from this manuscript are based on results using mitotic blockers including nocodazole and PLK1i. There is no evidence that phosphorylation of CEP55 by PLK1 itself plays a role in the process.

1st Revision - authors' response

27 April 2018

Replies to the editor and reviewer's comments

Editor

Although the referees find the study to be of potential interest, they also raise a number of significant concerns about the limited mechanistic insights provided (ref.1) and lack of rescue experiments (ref.2). Both referees make detailed comments and we would like to encourage you to address them all to the best of your availabilities. Please note that during our cross-commenting exercise, technical concerns emerged and the contribution of p53 in this setting should be better determined.

Response: Thank you for giving an opportunity to revise our manuscript. We have now provided new data related to reviewer's comments and addressed each point in detail. Regarding comment on limited mechanistic insights, please see our response to the reviewer comment below. Regarding the role of P53-CEP55 axis, Chang et al. 2012 has published detailed analyses on how P53 controls CEP55 levels (Chang et al, 2012), hence in this manuscript we have decided not to pursue this analysis in detail due to redundancy and time constraints to generate double knockdown (KD) of P53 and CEP55 cell lines in MCF10A. Moreover, in this manuscript we only use the P53 KD cells as a positive control for few experiments. In addition, we have separate manuscript (in preparation) showing data related to CEP55 overexpression in mouse models in conjunction with heterozygous P53 loss. We found that heterozygous CEP55 gain is sufficient to accelerate tumour formation in heterozygous p53 knockout mouse model, see figure below. For this reason, we believe the p53 data (old version Fig S2M) used previously as positive control in this manuscript is more appropriate to include in our P53-CEP55 related manuscript and validation will be done to assess CEP55 contribution in p53 settings.

Reviewer's comments

Referee #1 (Comments on Novelty/Model System for Author):

1. The technical quality is medium because they could have made more effort to deepen molecular signalling that control CEP55 or linking CEP55 to molecular pathways driving breast cancer subtypes.

Response: Thank you for this comment. A previous report by Chang et al. 2012 has published detailed analyses on how P53 negatively controls CEP55 levels (Chang et al, 2012) and in this manuscript we have provided evidence for the first time that CEP55 levels are also transcriptionally controlled by MYC oncogene and have focused our effort on this newly discovered regulation. Regarding linking CEP55 to molecular pathways driving breast cancer subtypes, CEP55 is overexpressed in all breast cancer subtypes compared to normal mammary epithelial cells, however, amongst subtypes basal like breast cancers show the highest expression shown using publically available clinical samples as well data on a larger panel of breast cancer cell lines (*Figure S1 and Figure 1A*). CEP55 is part of CIN70 signature, which drives aneuploidy and genomic instability in multiple cancers (Carter et al, 2006). Moreover, Birkbak et al. showed that amongst the breast cancer subtypes, basal-like breast cancers displayed the highest chromosomal structural complexity and chromosomal numerical instability defined through CIN70 score (Birkbak et al, 2011). Therefore, it is quite plausible that CIN70 genes, which includes other mitotic regulators (PLK1 and TPX2) drive this tumour subtype as a class effect (we have amended the text to reflect this). We also believe that CEP55 overexpression significantly contributes to accelerate tumour formation *in vivo* evident from our genetically engineered mouse model as shown above (unpublished data and the manuscript is in preparation). Notably, in this manuscript, we clearly illustrate the requirement of CEP55 in tumour initiation as near complete depletion of CEP55 in MDA-MB-231 cells abrogate tumour initiation *in vivo* (see *figure 1G*).

2. Novelty is low because there are previous papers (some of them by the authors themselves) describing CEP55 roles in aneuploidy.

Response: Although the role of CEP55 in the regulation of cytokinesis was first reported by us in 2005 (Fabro et al, 2005), many gene signature articles more recently have shown that CEP55 is part of CIN signature and the high CEP55 expression is correlated with tumor aneuploidy. Consistent with this, we previously provided evidence for fine-tune regulation of CEP55 expression where both overexpression and under expression can lead to cytokinesis failure. The novelty of this manuscript is i) we showed for the first time that in fact, unlike other mitotic proteins, CEP55 is required for aneuploid cell survival; ii) we showed that CEP55 is under control of MAPK signalling through MYC regulation; iii) Targeting MAPK signalling through MEK1/2 inhibitors markedly reduced CEP55 expression; hence it could be therapeutically targeted. Moreover, for the first time we mechanistically showed that CEP55-depleted cells enter mitosis prematurely in the presence of anti-mitotic agents. We also showed that CEP55 knockdown cells are sensitive to anti-mitotic agents particularly to PLK1 inhibition. Thus, combined PLK1 and MEK1/2 inhibition markedly reduced tumour formation in multiple xenograft models of basal breast cancers that overexpress CEP55. Our

study thus offers a novel strategy for exploiting this pathway in aggressive basal-like, TNBC. In our opinion, we have provided a complete translational preclinical data package for breast cancer therapy.

3. The study show that loss of CEP55 could sensitize breast cancer cells to anti-mitotic agents, suggesting further putative therapeutic avenues to treat breast cancer. Although the findings of the manuscript are interesting as well as their potential in breast cancer therapy however the weak of the manuscript relies on the loss of mechanistic novel insights. Indeed, in reviewer's eye the manuscript does not provide substantial novel insights about the molecular mechanisms involving CEP55, p53, Plk1 and MAP-kinase cascade. The above mentioned has been the main concern that reviewer could raise about the manuscript.

Response: Thank you for your comments. As mentioned before, we have provided data related to mechanism on how CEP55 expression is controlled by MAPK-MYC signalling (novel in this manuscript). We found that inhibition of MEK1/2 reduced mRNA and protein levels of CEP55 and we also provided evidence that MEK1/2 inhibitor's effect was predominantly mediated through the transcription factor MYC. Thus for therapeutic targeting of this pathway we have exploited MEK inhibitor to down regulate CEP55 expression and marked sensitivity of CEP55-downregulated cells, in terms of significant induction of cell death, to agents that perturb mitosis. The combination treatment caused synergistic cell death in control but not CEP55-depleted cells, through faster G2/M entry due to premature CDK activation validated through quantitative time-lapse imaging to measure time to and in mitosis (see additional data provided as *Figure EV5F-H*) and CDK activity assay. Mechanistic insights for targeting of CEP55-dependent tumours by combined MEK and PLK1 were further investigated through analysis of various apoptotic regulators. Also, in response to reviewer 2's-comment number 5, we have validated the mechanistic underpinning by performing converse experiment in CEP55 overexpressing MCF10A cells. The additional data has been included as Figure S5J-M in revised version.

Regarding the CEP55-P53 axis, see our response to the editor and our unpublished data on CEP55 genetically engineered mouse model. Moreover, apart from our original seminal findings (Fabbro et al, 2005), Chang et al, has recently provided direct link between CEP55-PLK1 axis which was negatively controlled by P53 (Chang et al, 2012). Therefore, the links between these proteins are well established in the literature in recent years (Bastos & Barr, 2010; Kamranvar et al, 2016; St-Denis et al, 2015).

4. The minor concerns are about the description of experimental procedures: although the technical quality of the manuscript is good, however sometimes it is very difficult to understand the experimental design and the reasons of some experimental choices.
In particular:

I) It is true for the first part of the manuscript running from the row 109 to 275. Authors should explain what their main objectives are and how they decide to pursue them. This could make the experimental procedures more understandable to readers.

Response: We have now amended the text from row 109-275 to reflect this point. See changes in red fonts, *page 7-12*.

II) Moreover authors should explain the reason why they use a "leaky" system to induce CEP55-depletion and what is the meaning of the different results they have obtained. (row 148-176)
"The inducible knockdown system used in this study was 'leaky' as evidenced by reduced CEP55 expression in both sh#2 and sh#8 polyclonal lines in the absence of doxycycline compared to scrambled shRNA-transfected cells", however it is not clear the reason why they use this leaky system.

Response: We previously stated the reason for using this line, see *page 9, line 157-168*. We also provide new data in the revision (Fig EV2G), showing that doxycycline-induced complete knockdown of CEP55 causes cytokinesis failure and multinucleation. Therefore, to minimize the pronounced negative impact of cytokinesis failure and multinucleation, we used the leaky cell lines, which express residual amount of CEP55 and fail to show cytokinesis defect to study the potential role of CEP55 in breast cancer. Moreover, as a comparison, we have now used Hs578T cell line

with constitutive knockdown of shCEP55.

III) Authors should explain why "breast cancer cells partially-depleted of CEP55 showed no significant changes in the number of cells displaying cytokinesis failure. In contrast near complete depletion of CEP55 resulted in a significant increase in multinucleated cells (data not shown)" Are those findings dependent on a pleiotropic effect of CEP55? After complete depletion of CEP55 multinucleated cells generated, are multinucleated cells living and aneuploid?

Response: Our data suggests that cytokinesis can proceed normally in the presence of residual small amount of CEP55, however doxycycline-induced complete loss of CEP55 leads to cytokinesis failure and multinucleation. We have added a new *figure EV2G*, in which we have cultured sh#2 and sh#8 clones in presence of doxycycline for 30 days, we do see time-dependent increase in multinucleation and cell death, suggesting that the multinucleated cells generated are losing viability. This statement has been added in the revised manuscript, *see page 9: line 157-161*.

Appended as Fig EV2G

IV) At row 182 authors say that p53 depletion is also necessary in order to make MCF10A-CEP55+, able to form tumor in nude mice, however to make the experimental procedure more clear to readers, authors should explain the reasons of this choice.

Response: We have changed the sentence to make it clear. We did not mention that p53 loss is required for MCF10A-CEP55 positive cells to form tumour. We simply use the p53 knockdown cells as a positive control in our experiments. We have now removed the data related to p53. Please see our response to the editor regarding contribution of p53.

5- Authors tested whether the CEP55-overexpressing MCF10A cells had acquired an oncogenic phenotype by measuring their ability to form colonies in non-adherent conditions and form acinar structures in Matrigel™. However it should be more appropriately referred to colonies in matrigel as mammospheres or spheroids. If cell colonies are generated after divisions of aneuploid cells, they will be, more probably, undifferentiated and unable to generate acini. Indeed what authors show in figure 1K are not acini rather spheroid colonies of cells exhibiting anchorage-independent growth. Moreover authors do not provide an immunofluorescence analysis of luminal markers proving that cell colonies are acini.

Response: We have now repeated the experiment twice showing that in fact a main cytoskeleton protein of epithelial cells, cytokeratin 19, staining intensity is reduced in CEP55 overexpressing MCF10A, see figure below. In fact as expected the empty vector transfected MCF10A cells formed hollow lumen compared to CEP55-overexpressing cells. To address the reviewer's comment, we have now changed the term from acinar structures to 'acinus-like spheroids', *see page 10: line 182-193*.

Appended as Fig 1K.

Label: Blue:DAPI; Red: Cytokeratin 19 and Green: Phalloidin.

6- Authors say that CEP55 depletion both in HeLa (previous published data) and MDA-MB-231 reduced aneuploid cell subpopulation however they also say that CEP55-depletion generated multinucleated cells. How could they explain that result?

Response: CEP55 depletion initially does generate multinucleated cells, however multinucleated cells generated are not viable long-term and are selected against the diploid cells in culture.

Referee #2 (Comments on Novelty/Model System for Author):

In this manuscript, Kalimutho and colleagues described the characterization of the effects of CEP55 expression in breast cancer. By analysing public databases, the authors found that high expression of CEP55 mRNA is associated with poor clinical outcomes in breast cancer. They also found that CEP55 is highly expressed in basal-like breast cancer cell lines. The authors then performed a series of experiments to study the effects of CEP55 downregulation and overexpression on cell growth, genome instability, and sensitivity to anti-mitotic drugs. They also found that inhibitors of MEK1/2 reduced the mRNA and protein expression of CEP55. Evidence including using siRNA against MYC indicated that the MEK1/2i effect may be acting through the transcription factor MYC. The authors then demonstrated that combining a MEK1/2i (AZD6244) and a PLK1i (BI2535) sensitized control cells but not partially CEP55-depleted cells. Finally, mouse xenograft models were also sensitized by MEK1/2i and PLK1i.

CEP55 is highly expressed in many cancers and is associated with chromosomal instability. Understanding its role in breast cancer and exploiting this pathway in cancer therapies are important issues. In general, the results are clear and the experiments are well-designed. However, a number of issues reduce my enthusiasm and I am not convinced by some of the authors' claims. These include the lack of rescue experiments and some converse CEP55-overexpressing experiments. The manuscript attempts to cover a wide range of topics (some thinly), hence some results are rather preliminary in nature and require additional investigation.

Response: We thank the reviewer for recognising the importance of this study. It is true that this manuscript has tried to cover large number of experiments and some of them are underdeveloped. We have now removed data related to p53 as we are currently preparing a manuscript related to p53 using engineered mouse model of Cep55 overexpression to study its role in tumorigenesis as described above. We believe our previous data on p53-mediated regulation of Cep55 expression is more appropriate for our p53 related manuscript (see our response to the editor).

Points

1. The authors mainly used two clones of MDA-MB-231 expressing shRNA against CEP55 (using the leakage of the inducible promoter) to study the effects of CEP55 downregulation.

Response: As already shown in Fig. S3, we also used CEP55-knockdown Hs578T line in a few limited experiments. We have now generated more data using these lines to confirm universality of the mechanism involved.

Appended as Fig EV4J

Appended as Fig EV5I

2. In some experiments, the difference in expression of CEP55 between clone#2 and #8 is not so obvious as that shown in Fig 1D (e.g. Fig S2F). The authors should quantify the percentage of knockdown in these two clones relative to the parental cells (e.g. using Western blots with standard curves).

Response: We have repeated this experiment and quantified the base line CEP55 expression levels as shown below, we consistently see ~50% reduction in CEP55 levels in sh#2 and 80% reduction in sh #8 transfected cells.

Appended as Fig EV2F

3. Furthermore, referring these cells as "CEP55-depleted cells" in the manuscript is misleading.

Response: We have changed the term 'depletion' to 'knockdown' throughout the manuscript.

-Another important information appears to be missing is whether is the level of CEP55 after downregulation with shRNA was comparable to cells with low CEP55 (MCF-10A)?

Conversely, the authors generated MCF-10A clones overexpressing CEP55 (Fig 1H). Is the level of the exogenous CEP55 comparable to the endogenous CEP55 found in other cell lines that normally express CEP55 (such as MDA-MB-231)?

Response: We have now run a panel of cell lines together to show that the level of ectopic CEP55 expressed in MCF10A is comparable to endogenous CEP55 found in highly aggressive MDA-MB-231 cells and the knockdown achieved in MDA-MB-231 sh#8 is comparable to endogenous CEP55 level seen in MCF10A cells. We have amended the manuscript to reflect this this point, *page 9: line 179-180*.

CEP55 expression across different cell lines.

4. A major shortcoming of this manuscript is the majority of the experiments were performed using the two clones of MDA-MB-231 expressing CEP55 shRNA. No rescue experiments (re-introducing CEP55) were performed.

Response: We have now introduced representative rescue experiments using shRNA resistant construct as shown below. The data related to rescue experiments have been incorporated in the revised version and discussed accordingly, *page 9:line 165-167; page 15:314-317; page 16:line 350-355; page 21:line 460-461*.

Appended as Fig EV2H, I.

Appended as Fig EV4H.

Appended as Fig EV5F-H.

Appended as Fig EV8E.

5. The authors demonstrated CEP55 downregulation resulted in reduced proliferation, anchorage-independent colony formation, migration and invasion, and increased chromosomal stability. Some experiments were then performed using CEP55-overexpressing cells to consolidate the findings (e.g. overexpression of CEP55 induced chromosomal instability of chromosome 20). Why CEP55 overexpression experiments were not performed for other growth characteristics?

Response: We did do those experiments previously and the results are shown in figure 1 and supplementary figures, see Fig. 1 panel I, J, K and Fig. S2N, O in the revised version.

(5) The authors found that partially CEP55-depleted cells entered G2/M faster after incubation with nocodazole or PLK1i (but not in unperturbed cells). The data are intriguing but not very convincing. The authors should use live-cell imaging to measure the time of mitosis carefully. Again, converse experiments using CEP55-overexpressing cells will be useful.

Response: We have now provided new data related to this. We also rescued the phenotype by introducing shRNA-resistant CEP55 expression construct in CEP55 knockdown MDA-MB-231, page 16:line 350-355; page 21.

Appended as Fig EV5F-H.

Also, we have included data related in CEP55-overexpressing MCF10A cells in the revised version of the manuscript, page 17:line 360-362.

Appended as Fig EV5F-H.

The comment of the lack of experiments using CEP55-overexpressing cells also apply to when the authors examine the effects of combined MEK1/2i and PLK1i treatment.

Response: We have now provided new data related to this concern using both MDA-231 (for rescue phenotype, see Figure S 8E) and CEP55-overexpressing MCF10A cells. We only see a marginal increase in apoptosis in CEP55-overexpressing MCF10A cells with combination of MEK and PLK inhibitor compared to cancer lines expressing high endogenous levels of CEP55, page 21:line 462-465.

Appended as Fig EV8F.

(6) The authors showed that depletion of p53 in MCF10A increased CEP55 expression. However, they are largely preliminary in nature and require additional investigation to reveal potential functional links between p53 and CEP55. For example, the authors should examine the contribution of the increase in CEP55 in the cellular effects of p53 depletion.

Response: As mentioned previously, we have removed data related to p53. See our above response to the editor.

(7) The authors believed that the apoptosis during mitosis in partially CEP55-depleted cells involved BCL-XL and BIM. These data are too preliminary for the authors to draw this conclusion. Converse experiments of CEP55 overexpression were also not performed.

Response: Thank you for your comments. We have now performed this analysis in two additional cell lines- CEP55 knock-down Hs578T and CEP55 overexpressing MCF10A cells. Although we do observe consistent PARP and Caspase 3 cleavages across the two lines in response to nocodazole treatment in a manner rescued by CEP55 expression construct, the effects on other apoptotic proteins (BCL2 and MCL1) were not consistent between MDA-MB-231, HS578T and MCF10A cells. We observed that BCL-XL and BIM were impacted across these lines. Therefore, we also performed siRNA-mediated knockdown against BCL-XL and BIM and found that BCL-XL knockdown enhanced apoptosis in control MDA-MB-231 cells after nocodazole treatment. However, knockdown of BIM failed to rescue nocodazole-induced apoptosis in CEP55 knockdown cells. Further detailed analyses are required to delineate CEP55-dependent regulation of BCL2-family members. However, this is outside of the scope of the current study. We have added these findings into supplementary data, and amended the manuscript accordingly to reflect these points, page 15-16:306-336.

Appended as Fig EV4H, I, J and K.

(8) The meaning of error bars and the number of samples are often missing in the Figure legends (e.g. 2A, 2D, 2F, 3B, 3C, 3F, 3H etc.). Where indicated, they are not very consistent (e.g. the end of the legends of Fig 1 indicates n=2-3; but this probably is not true for part G). Furthermore, some of the experiments with n=2 need more repeats to justify using error bars).

Response: We have now repeated some of the experiments and have amended the expanded view legends accordingly.

(9) Fig 7F: in the model, the authors indicate a role of PLK1 in controlling CEP55 functions. However, conclusions from this manuscript are based on results using mitotic blockers including nocodazole and PLK1i. There is no evidence that phosphorylation of CEP55 by PLK1 itself plays a role in the process.

Response: We agree with the reviewer on this. We have now removed the phosphorylation and modified the figure to reflect his/her point.

Fig 7F

References

- Bastos RN, Barr FA (2010) Plk1 negatively regulates Cep55 recruitment to the midbody to ensure orderly abscission. *J Cell Biol* 191: 751-760
- Birkbak NJ, Eklund AC, Li Q, McClelland SE, Endesfelder D, Tan P, Tan IB, Richardson AL, Szallasi Z, Swanton C (2011) Paradoxical relationship between chromosomal instability and survival outcome in cancer. *Cancer Res* 71: 3447-3452
- Carter SL, Eklund AC, Kohane IS, Harris LN, Szallasi Z (2006) A signature of chromosomal instability inferred from gene expression profiles predicts clinical outcome in multiple human cancers. *Nat Genet* 38: 1043-1048
- Chang YC, Wu CH, Yen TC, Ouyang P (2012) Centrosomal protein 55 (Cep55) stability is negatively regulated by p53 protein through Polo-like kinase 1 (Plk1). *J Biol Chem* 287: 4376-4385
- Fabbro M, Zhou BB, Takahashi M, Sarcevic B, Lal P, Graham ME, Gabrielli BG, Robinson PJ, Nigg EA, Ono Y et al (2005) Cdk1/Erk2- and Plk1-dependent phosphorylation of a centrosome protein, Cep55, is required for its recruitment to midbody and cytokinesis. *Dev Cell* 9: 477-488
- Kamranvar SA, Gupta DK, Huang Y, Gupta RK, Johansson S (2016) Integrin signaling via FAK-Src controls cytokinetic abscission by decelerating PLK1 degradation and subsequent recruitment of CEP55 at the midbody. *Oncotarget* 7: 30820-30830
- St-Denis N, Gupta GD, Lin ZY, Gonzalez-Badillo B, Pelletier L, Gingras AC (2015) Myotubularin-related proteins 3 and 4 interact with polo-like kinase 1 and centrosomal protein of 55 kDa to ensure proper abscission. *Mol Cell Proteomics* 14: 946-960

2nd Editorial Decision

15 June 2018

Thank you for the submission of your revised manuscript to EMBO Molecular Medicine. We have now received the enclosed report. As you will see, the reviewer is supportive and I am pleased to inform you that we will be able to accept your manuscript pending minor editorial amendments.

I look forward to reading a new revised version of your manuscript as soon as possible.

***** Reviewer's comments *****

Referee #1 (Comments on Novelty/Model System for Author):

As already described in in the previous review, the main concern about the manuscript is still dependent on the loss of mechanistic novel insights, point that has been addressed by authors by

citations of previous published evidences. However, if editor agrees, this manuscript could be considered suitable for publication if we look at its potential therapeutic implication as it is suggesting a new combination therapy for TNBC.

Referee #1 (Remarks for Author):

The mechanism of how CEP55 mediates genomic instability and tumorigenesis is unclear yet, and, as previously highlighted by reviewer, there are lots of known in the story: correlation of CEP55 with survival and clinical outcome, its oncogenic role and regulation by p53. We also know that depletion of CEP55 using siRNAs in Hela cells resulted in cytokinesis failure leading to multinucleation (Fabbro et al, 2005).

Moreover, mechanistic insights, as regulation through MAPK and MYC, is quite correlative and some of them expected I think.

Conversely, what is novel is the reduced aneuploid cell population as well as CNA after CEP55 depletion in BC, shown in vitro and confirmed in patient data, supporting the hypothesis that CEP5-depleted cells are sensitive to agents that perturb mitosis, like PLK-1, and undergo apoptosis through premature CDK1 activation and SAC-dependent mitotic catastrophe.

As a consequence, the combined treatment MEK1/2-PLK1i is clinically interesting and they have good in vivo data.

Therefore, I would consider this manuscript for its potential clinical implication as they are suggesting a new combination therapy for TNBC.

Corresponding Author Name: Murugan Kalimutho, Kum Kum Khanna

Manuscript Number: EMM-2017-08566